# A natural *timeless* polymorphism allowing circadian clock synchronization in "white nights"

Angelique Lamaze [1✉], Chenghao Chen[2,3✉], Solene Leleux[1], Min Xu[3], Rebekah George[1] & Ralf Stanewsky [1✉]

Daily temporal organisation offers a fitness advantage and is determined by an interplay between environmental rhythms and circadian clocks. While light:dark cycles robustly synchronise circadian clocks, it is not clear how animals experiencing only weak environmental cues deal with this problem. Like humans, Drosophila originate in sub-Saharan Africa and spread North up to the polar circle, experiencing long summer days or even constant light (LL). LL disrupts clock function, due to constant activation of CRYPTOCHROME, which induces degradation of the clock protein TIMELESS (TIM), but temperature cycles are able to overcome these deleterious effects of LL. We show here that for this to occur a recently evolved natural timeless allele (*ls-tim*) is required, encoding the less light-sensitive L-TIM in addition to S-TIM, the only form encoded by the ancient *s-tim* allele. We show that only *ls-tim* flies can synchronise their behaviour to semi-natural conditions typical for Northern European summers, suggesting that this functional gain is driving the Northward *ls-tim* spread.

---

[1] Institute of Neuro- and Behavioral Biology, Westfälische Wilhelms University, Münster, Germany. [2] Department of Physiology and Biophysics, University of Washington, Seattle, WA, USA. [3] Howard Hughes Medical Institute, Janelia Research Campus, Ashburn, VA, USA. ✉email: alamaze@uni-muenster.de; chenc@janelia.hhmi.org; stanewsky@uni-muenster.de

Like most organisms, *Drosophila melanogaster* rely on their endogenous circadian clock to regulate rhythmic physiological and behavioural outputs. This timer is equipped with two core clock proteins CLOCK(CLK) and CYCLE(CYC) to activate the transcription of the clock genes *period* (*per*) and *timeless* (*tim*). The translated PER and TIM proteins then terminate their own transcription through negative feedback[1]. This transcription/translation feedback loop, constitutes the molecular oscillator of the biological clock, which runs with a period of ~24 h, even in absence of environment cues. On the other hand, this robust timing system interacts with the environment and resets itself by daily cues like fluctuating light and temperature (so called "Zeitgeber"). CRY is an important blue light photoreceptor expressed in the *Drosophila* eye as well as in subsets of the clock neurons, which are composed of about 75 neurons expressing core clock genes in each brain hemisphere[2–4]. This central pacemaker contains seven anatomically well-defined clusters: three groups of dorsal neurons (DN1-3), the lateral posterior neurons (LPN), the dorsal lateral neurons (LNd) and the large and small ventral lateral neurons (l- and s-LNv). Together, they orchestrate timing of the locomotor activity patterns with external light and temperature fluctuations. When flies are exposed to light, CRY is activated and binds to TIM and the F-box protein JETLAG (JET), triggering TIM and CRY degradation in the proteasome to reset the clock network[5–8]. Therefore, exposure of flies to constant light (LL) leads to arrhythmicity, due to the constitutive degradation of TIM in clock neurons, mediated by CRY[9,10]. In addition, rhodopsin-mediated retinal photoreception contributes to circadian light input, and only if both CRY and the visual system function are ablated in parallel, circadian light synchronization is abolished[11]. Another important Zeitgeber to synchronise circadian rhythms is temperature. In mammals, temperature cycles (TC) with an amplitude of 1.5 °C induce robust circadian gene expression in cultured tissues[12]. Moreover, the daily fluctuation of body temperature (36–38.5 °C) generated by the suprachiasmatic nucleus (SCN) is employed to enhance internal circadian synchronization[13]. In *Drosophila*, unlike cell autonomous light resetting by CRY, clock neurons receive temperature signals from peripheral thermo sensory organs including the aristae and mechanosensory chordotonal organs[14–17]. Interestingly, robust molecular and behavioural entrainment to temperature cycles was observed under LL[18,19], suggesting that cycling temperature can somehow rescue clock neurons from the effects of constant light, but the underlying molecular mechanism is unknown.

This ability to synchronise circadian clocks to temperature cycles in constant light may have ecological relevance. For instance, animals living above or near the Northern Arctic Circle experience LL or near-LL conditions, while the temperature still varies between "day" and "night" (due to differences in light intensity). In Northern Finland summers (e.g., Oulu, 65° North), the sun only sets just below the horizon and it never gets completely dark, so that organisms experience so called "white nights". At the same time, average temperatures vary by 10 °C between day and night (www.timeanddate.com/sun/finland/oulu?month=7&year=2021), suggesting that animals use this temperature difference to synchronise their circadian clock. *Drosophila melanogaster* populate this region, with massive expansion of the population during the late summer. It has been suggested that a recently evolved novel allele of the *tim* gene is advantageous for Northern populations and that this allele is under directional natural selection[20–22]. The novel *ls-tim* allele encodes a longer (by 23 N-terminal amino acids), less-light sensitive form of TIM (L-TIM) in addition to the shorter (S-TIM) form, the only form encoded by the ancient *s-tim* allele[7,8,20,23]. The reduced light-sensitivity of L-TIM is caused by a weaker light-dependent interaction with CRY, thereby resulting in increased stability of L-TIM during light, compared to S-TIM[7,8,20]. Indeed, *ls-tim* flies show reduced behavioural phase-responses to light pulses[20] and are more prone to enter diapause during long summer days compared to *s-tim* flies[21]. It has been proposed that light-sensitivity of circadian clocks needs to be reduced in Northern latitudes, in order to compensate for the long summer days and presumably excessive light reaching the clock cells[24]. The *ls-tim* allele might therefore offer a selective advantage in Northern latitudes, which is indeed supported by the spread of this allele from its origin in Southern Italy 300—3000 years ago by directional selection[21,22].

Here we provide strong support for this idea, by showing that only *ls-tim* flies are able to synchronise their circadian clock and behavioural rhythms to temperature cycles in constant light (LLTC). The observation that wild type flies carrying the ancient *s-tim* allele are not able to synchronise to LLTC support the advantage of the *ls-tim* allele in Northern latitudes. Despite of their reduced light sensitivity, *ls-tim* flies can still synchronise their circadian clock because they can use temperature cycles as Zeitgeber. We show that *ls-tim* is also required for synchronisation under semi-natural conditions mimicking "white nights" conditions as they occur in natural Northern latitude habitats of *Drosophila melanogaster*, supporting the adaptive advantage of this allele.

## Results

**ls-tim, but not s-tim flies are able to synchronise to temperature cycles in constant light.** During our studies of how temperature cycles synchronise the circadian clock of *Drosophila melanogaster*, we noticed that some genetic control stocks did not, or only very poorly, synchronise their behavioural rhythms to temperature cycles in constant light (16 °C: 25 °C in LL). Further analysis revealed that the ability to synchronise to LLTC was correlated with the presence of the *ls-tim* allele, while flies that did not, or only poorly synchronise, carry the *s-tim* allele. This was true for the two isogenic strains iso31 (*ls-tim*)[25] and iso (*s-tim*)[26] (Fig. 1a-c). While *ls-tim* flies show a synchronized evening peak in the second half of the warm phase, *s-tim* flies lack this activity peak and only increase their activity briefly after the cold transition (Fig. 1a, b). To quantify this synchronisation, we also calculated the slope of the activity increase leading up to the evening peak between the lowest and highest median time point of the normalized activity for each genotype and individual (see Methods). Using this method, we observed a slope of zero (Δactivity/Δt) for *s-tim* while *ls-tim* have a positive slope, indicating normal synchronization to LLTC (Fig. 1c). To test if the presence of the *ls-tim* allele is indeed required for synchronisation to LLTC, we analysed additional control and wild type strains, either from lab collections (the widely used *y* w and Canton-S stocks,) or from stocks recently collected in the wild (Fig. S1, S2). Interestingly, all stocks carrying *ls-tim* showed a synchronized activity peak in the second half of the warm phase during LLTC, similar as described above for iso31 (Fig. S1, S2). In contrast, all stocks carrying the *s-tim* allele only responded to the temperature decrease, or showed a very broad activity peak lasting for the majority of the warm phase (Fig. S1, S2). Furthermore, as expected for wild type flies, independent of *s-tim* or *ls-tim*, all control stocks became arrhythmic in LL at constant temperatures (Fig. 1a, S1a, S2a)[27]. Finally, when exposed to temperature cycles in DD, both *s-tim* and *ls-tim* robustly synchronised their behavioural activity (Fig. S1c), indicating that the *s-tim* allele specifically affects clock synchronization during LLTC.

To test if the early activity increase observed in some of *s-tim* strains reflects proper synchronisation to LLTC, we compared

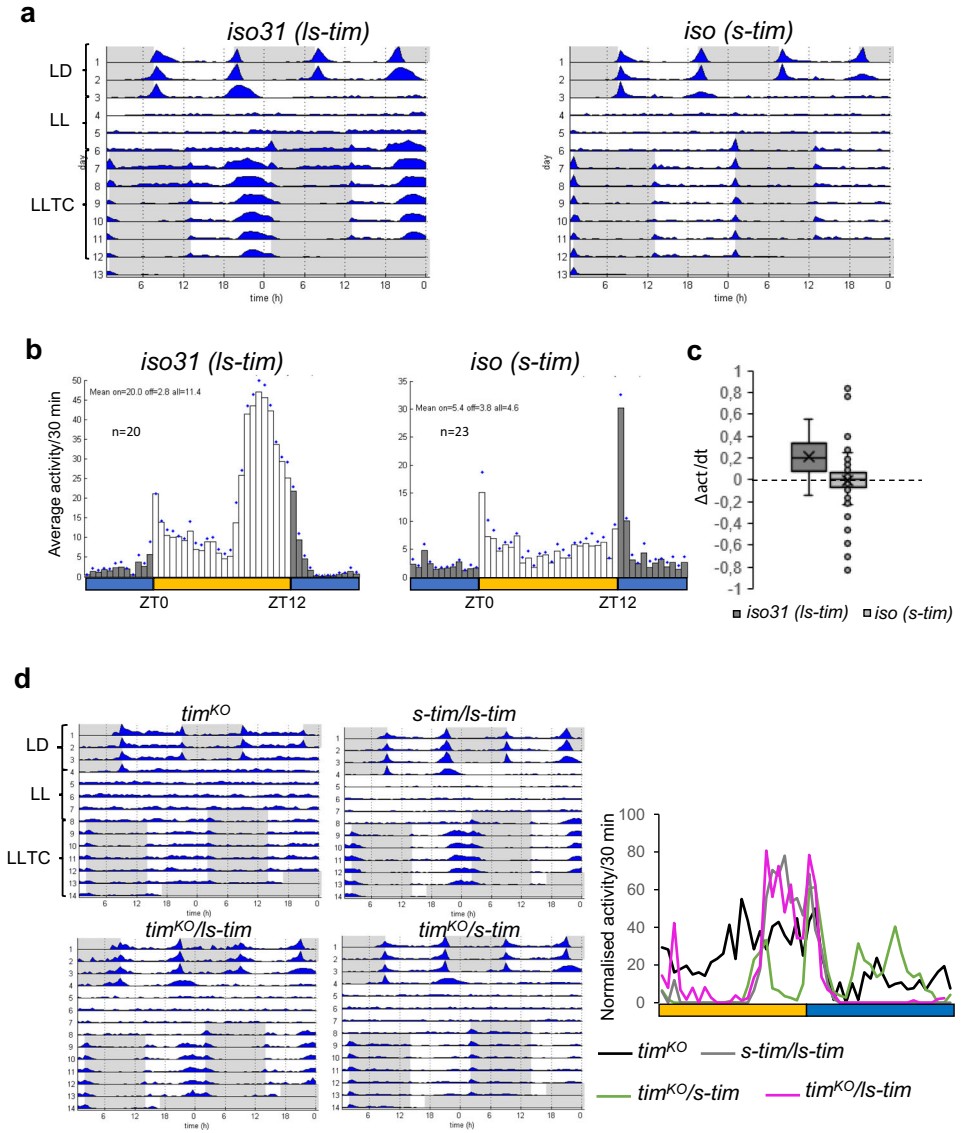

**Fig. 1 *s-tim* flies cannot synchronise their behaviour to temperature cycles in constant light. a** Group actograms of the genotypes indicated representing average activity of one experiment. Environmental conditions are indicated next to the actogram on the upper left. LD: 12h-12h Light-Dark constant 25 °C, LL: constant light and 25 °C, LLTC: LL and 25–16 °C temperature cycles. White areas: lights-on, and 25 °C, grey areas: lights-off and 25 °C during LD and lights-on and 16 °C during LLTC. N (*w, iso31 ls-tim*): 20, (*w, iso s-tim*): 23. **b** Histograms of the average activity of three consecutive LLTC (day 4 to 6). Same flies as in (**a**). Yellow bar: thermophase, blue bar cryophase (12 h each). Blue diamonds indicate Standard Error of the Mean (SEM). **c** Box plots showing the slope of the evening peak on the 6th day of LLTC (combined experiments) (see "Methods" for calculation). $ZT_{min}$(iso31) = 7, $ZT_{min}$(iso *s-tim*) = 9.5; $ZT_{max}$(iso31) = 10, $ZT_{max}$(iso *s-tim*) = 11.5. N (*w, iso31 ls-tim*): 74, (*w, iso s-tim*): 65. The lowest line of the box plot indicates the first interquartile, the centre line the median, the upper line the third interquartile, the cross the average, and the whiskers indicate the minimum and maximum except for out layers. **d** Left: Group actograms of the genotypes indicated representing average activity of one experiment as in (**a**). Right: median of the normalised activity during LLTC6. N: (*tim^KO*): 9, (*w;s-tim/ls-tim*): 15, (*tim^KO/ls-tim*): 5, (*tim^KO/s-tim*): 12. Source data are provided as a Source Data file.

behaviour of *s-tim* flies with *tim^KO* mutant flies. *tim^KO* is a new *tim* null allele in the *iso31* background[28]. Interestingly, although *tim^KO* flies do not show clock-controlled behaviour in LD, their locomotor activity pattern in LLTC is equivalent to what we observed in some *s-tim* stocks (Fig. 1d). We therefore conclude that *s-tim* flies are not able to synchronise their clock controlled behavioural activity rhythms to temperature cycles in constant light. Next, we compared the behaviour of hemizygous *s-tim* and *ls-tim* flies. While *tim^KO/ls-tim* flies showed normal LD and LLTC behaviour, *tim^KO/s-tim* flies only synchronised to LD (Fig. 1d). Interestingly, trans-heterozygous *s-tim/ls-tim* flies perfectly synchronise their behaviour to LLTC, showing that *ls-tim* is dominant over *s-tim* for LLTC entrainment (Fig. 1d).

**s-tim flies fail to properly synchronise their clock protein oscillations to temperature cycles in constant light.** To distinguish if the lack of behavioural synchronization is due to a defect within or downstream of the circadian clock, we analysed PER and TIM oscillations during LLTC in clock neurons of *s-tim* and *ls-tim* flies (Fig. 2, S3). As expected for *s-tim* in LL, TIM levels were lower compared to *ls-tim* flies, but detectable at all four time points we examined (ZT0, ZT6, ZT12, ZT18). Importantly, the amplitude of TIM oscillations in *s-tim* was dramatically reduced compared to *ls-tim* flies. Moreover, in most of the ventral and dorsal lateral clock neurons and potentially in the DN2, TIM oscillations in *s-tim* are phase advanced by 6 h, reaching peak values at ZT6 compared to ZT12 for *ls-tim*, while in the s-LNv S-

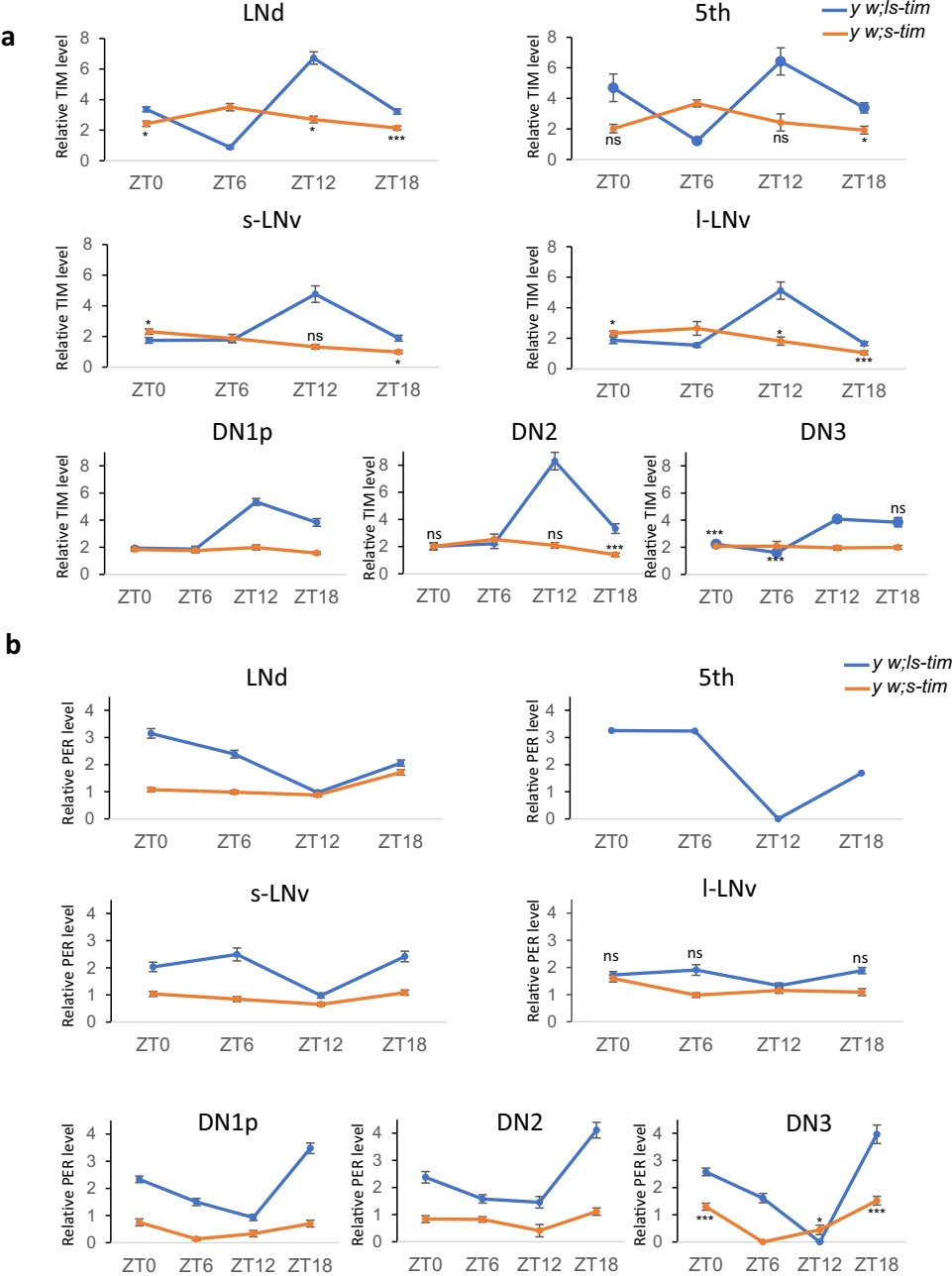

**Fig. 2 During constant light and temperature cycles TIM and PER oscillations are strongly dampened in clock neurons of *s-tim* flies.** Averages of normalised TIM (**a**) and PER (**b**) levels (see "Methods") in the different groups of clock neurons during day six of LLTC in *y w; ls-tim* (blue lines) and *y w; s-tim* flies (orange lines). Number of brain hemispheres/time points = 4–5. Error bars = *sem* (standard error of the mean). PER was not detectable in the 5th s-LNv of *s-tim* flies. A Mann–Whitney test was performed (with a Bonferroni correction) to compare *s-tim* ZT6 with the other time points to determine the significance of potential oscillations. For TIM, no significant oscillations were observed in the DN1 and DN3 groups. For PER, only the DN3 showed significant oscillations. In *ls-tim* all time points in (**a**) and (**b**) differed significantly from ZT12 (peak for TIM and trough for PER) apart from the l-LNv (**b**): * $p < 0.05$, ** $p < 0.01$, *** $p < 0.0001$. Source data are provided as a Source Data file.

TIM peaked at ZT0 (Fig. 2a). In the DN1 and DN3, S-Tim levels were at constant low levels at all four time points (Fig. 2a). We also noticed that even in *ls-tim* flies, TIM peaks earlier compared to LD and constant temperature conditions[29], correlated with the phase advance of the behavioural evening peak in LLTC compared to LD (Fig. 1, S1, S2)[30]. In addition, we found that S-TIM remains cytoplasmic at all time points studied, while in *ls-tim* flies TIM showed the typical nuclear accumulation at ZT0 (Fig. S3a, c)[29]. Similarly, PER levels were drastically reduced in *s-tim* compared to *ls-tim* flies in the PDF-positive LNv and LNd, and

PER was undetectable in the 5th s-LNv (Fig. 2b, S3b, d). Due to the low levels it was impossible to clearly distinguish between cytoplasmic and nuclear localisation, but the results indicate constitutive nuclear and cytoplasmic PER distribution at all four time points examined (Fig. S3 b, d). The only exception were the DN3, which showed significant PER oscillations in *s-tim* flies ($p < 0.0001$, Mann–Whitney test), indicating the existence of an alternative system to control PER oscillations at least in this group of neurons (Fig. 2b). Overall, the results indicate that the drastic impairment of synchronised TIM and PER protein

expression in clock neurons underlies the inability of *s-tim* behavioural synchronization to LLTC.

**Cryptochrome depletion allows synchronisation of *s-tim* flies to temperature cycles in constant light**. *s-tim* flies are more sensitive to light compared to *ls-tim* flies, presumably because the light-dependent interaction between CRY and S-TIM is stronger compared to that of CRY and L-TIM[8,20]. To test if the inability of *s-tim* flies to synchronise to LLTC is due to the increased S-TIM:CRY interaction and subsequent degradation of TIM[8], we compared the behaviour of *s-tim* and *ls-tim* flies in the absence of *cry* function using the same environmental protocol. As expected, *cry[02]* mutant flies showed rhythmic behaviour in LL and constant temperature (Fig. 3a)[9,26]. Strikingly, the *s-tim* flies lacking CRY were now able to synchronise to LLTC, similar to *cry[02]* flies carrying the *ls-tim* allele (Fig. 3a, b). Also, the slope of the evening activity increase is positive for both *ls-tim* and *s-tim cry[02]* flies, confirming synchronization to LLTC for both genotypes (Fig. S4a). The slightly lower slope value in *s-tim cry[02]* flies on day 6 of LLTC ($p < 0.05$, Kruskal two-group comparison) indicates a partial rescue of *s-tim* by the removal of CRY function, although there is no difference on day two in constant conditions after the TC (Fig. S4a). Moreover, upon release into LL and constant temperature, activity peaks of both genotypes were aligned with those during the last few days in LLTC, indicating stable synchronisation of clock-driven behavioural rhythms (Fig. 3a, b).

**Cryptochrome depletion partially restores molecular synchronisation of *s-tim* flies to temperature cycles in constant light**. The behavioural results of *s-tim* flies lacking CRY described above, suggest that PER and TIM protein oscillations within clock neurons that underlie behavioural rhythms are also synchronised in LLTC. To confirm that *s-tim* flies lacking functional CRY are able to synchronise their molecular clock, we determined PER and TIM levels in different subsets of clock neurons of *s-tim cry[02]* flies at four different time points during LLTC. Overall, we observed robust TIM oscillations in Lateral and Dorsal clock neurons of *s-tim cry[02]* flies, demonstrating that removal of CRY restores molecular synchronisation in *s-tim* flies during LLTC (Fig. 3c, S4b, c). Similar results were obtained for PER, except that in the large and small LNv PER oscillated only with a weak, but significant amplitude ($p < 0.0001$, Mann–Whitney test) (Fig. 3d, S4b, c). Nevertheless, PER and TIM oscillations were not identical to those observed in *ls-tim cry[+]* flies under the same conditions (compare Fig. 3c, d with Fig. 2a, b). To our surprise, we found desynchronization between and within groups. (Fig. 3d, S4b, c). While the amplitude of PER and TIM oscillation is comparable within the LNd and 5th s-LNv, there was a clear phase difference between the CRY-positive LNd and the 5th s-LNv compared to the CRY-negative LNd (the LNd were distinguished based on the larger size of the CRY-positive neurons), with the trough of PER and TIM in the CRY-negative LNd phase-advanced by at least 6 h compared to the CRY-positive neurons (Fig. 3c, d, S4b). Moreover, the overall TIM phase in the CRY-negative LNd is advanced by 6 h compared to that of PER. Apart from half of the ~15 DN1p neurons and a few DN3, the neurons belonging to the three DN groups do not express CRY[4]. Interestingly, in these neurons TIM peaks at ZT12 as in the CRY-negative LNd, with the DN1p oscillating with the highest amplitude (Fig. 3c). To summarize, in *s-tim cry[02]* flies, the six LNd and the 5th s-LNv are the only clock neurons showing high amplitude PER oscillations, and the CRY-negative LNd, and DN neurons show drastic phase advances of PER (LNd only) and TIM oscillations compared to the CRY-positive LNd and 5th s-LNv evening cells.

**Rhodopsin photoreception contributes to circadian clock synchronization in constant light and temperature cycles**. The constitutive cytoplasmic localisation of TIM in *s-tim* flies during LLTC in both CRY-positive and CRY-negative cells (Fig. S3a, c), suggests that the visual system also contributes to circadian temperature synchronisation in the presence of light. To test this hypothesis, we analysed *s-tim* flies lacking CRY, in which Rhodopsin-expressing photoreceptor cells are either absent (via cell ablation using GMR-hid), or in which the major photo transduction cascade is interrupted due to the absence of Phospholipase C-ß (PLC-ß, via loss-of-function mutation of *norpA*). Completely removing both the visual system and CRY renders the brain clock blind to light entrainment[11], which is exactly what we observed with the *s-tim GMR-hid cry[01]* flies analysed here (Fig. 4a). In contrast, due to *norpA*-independent Rhodopsin photoreception, *norpA[P41] cry[b]* double mutants can still be entrained to LD[31–33], consistent with what we here observe for the *norpA[P41] s-tim cry[02]* double mutants (Fig. 4a). Strikingly, after switch to LLTC both genotypes synchronise their behaviour, however with a clear phase advance compared to *s-tim cry[02]* flies in the same condition (Fig. 4b). Interestingly, *norpA[P41] s-tim cry[02]* double mutants take longer to establish a similar early phase as the *s-tim GMR-hid cry[01]* flies (Fig. 4a). We attribute this difference to the initial synchronization of *norpA[P41] s-tim cry[02]* flies to the LD cycle, and their maintained synchronised free running activity in LL and constant temperature (Fig. 4a, b). In contrast, the *s-tim GMR-hid cry[01]* flies are completely desynchronised at the beginning of the LLTC, presumably allowing for rapid synchronisation to the temperature cycle. In conclusion, the results indicate that photoreceptors using a *norpA*-dependent signalling pathway play a role in phasing the behaviour in LLTC.

To see if the Rhodopsin contribution to phasing behaviour in LLTC has a molecular correlate, we analysed TIM expression in *s-tim* flies lacking PLC-ß and CRY (*norpA[P41] cry[02]*). Overall, we again observed robust TIM and PER oscillations in all clock neurons, very similar in phase and amplitude to the oscillations in *s-tim cry[02]* single mutants (Fig. 4c, S5, compare to Fig. 3c, d, S4). However, in the s-LNv the amplitude of PER oscillations was substantially increased in the absence of *norpA* function. Moreover, the phase difference of TIM between the CRY-positive and CRY-negative LNd observed in the *cry[02]* single mutants is reduced in the *norpA[P41] cry[02]* double mutants, because the CRY-positive neurons show an earlier increase of TIM levels (Fig. 4c). Nevertheless, we do not think these differences can explain the drastic phase advance of *cry[02]* mutants with a defective visual system. In contrast we favour the idea that a switch of the neuronal network balance causes the behavioural phase advance, because wild type flies show the same early activity phase when synchronized to temperature cycles in constant darkness instead of constant light (Fig. S1c)[30].

**ls-tim expression in clock neurons is sufficient for temperature synchronisation in constant light**. Because *norpA*-dependent visual system function contributes to synchronization of TIM oscillations in clock neurons during LLTC (Fig. 4c), we wondered if expressing the *ls-tim* allele specifically in clock neurons or photoreceptors in otherwise *s-tim* flies, would also restore synchronization. For this we first recombined a *UAS-ls-tim*[34] transgene with the *tim[KO]* allele (Methods) and crossed the recombinant flies to *s-tim* flies and *tim[KO]* stocks. As expected, *UAS-ls-tim, tim[KO] / s-tim* and *UAS-ls-tim, tim[KO] / tim[KO]* flies did not synchronise to LLTC (Fig. 5a, b, S6a). Next, we crossed *UAS-ls-tim, tim[KO]* flies to *Clk856-Gal4* (expressed in all clock neurons and not in photoreceptor cells)[35], and to *Rh1-Gal4* (expressed in photoreceptor cells R1 to R6, but not in clock neurons)[36]. Strikingly, expression of *ls-tim* in clock

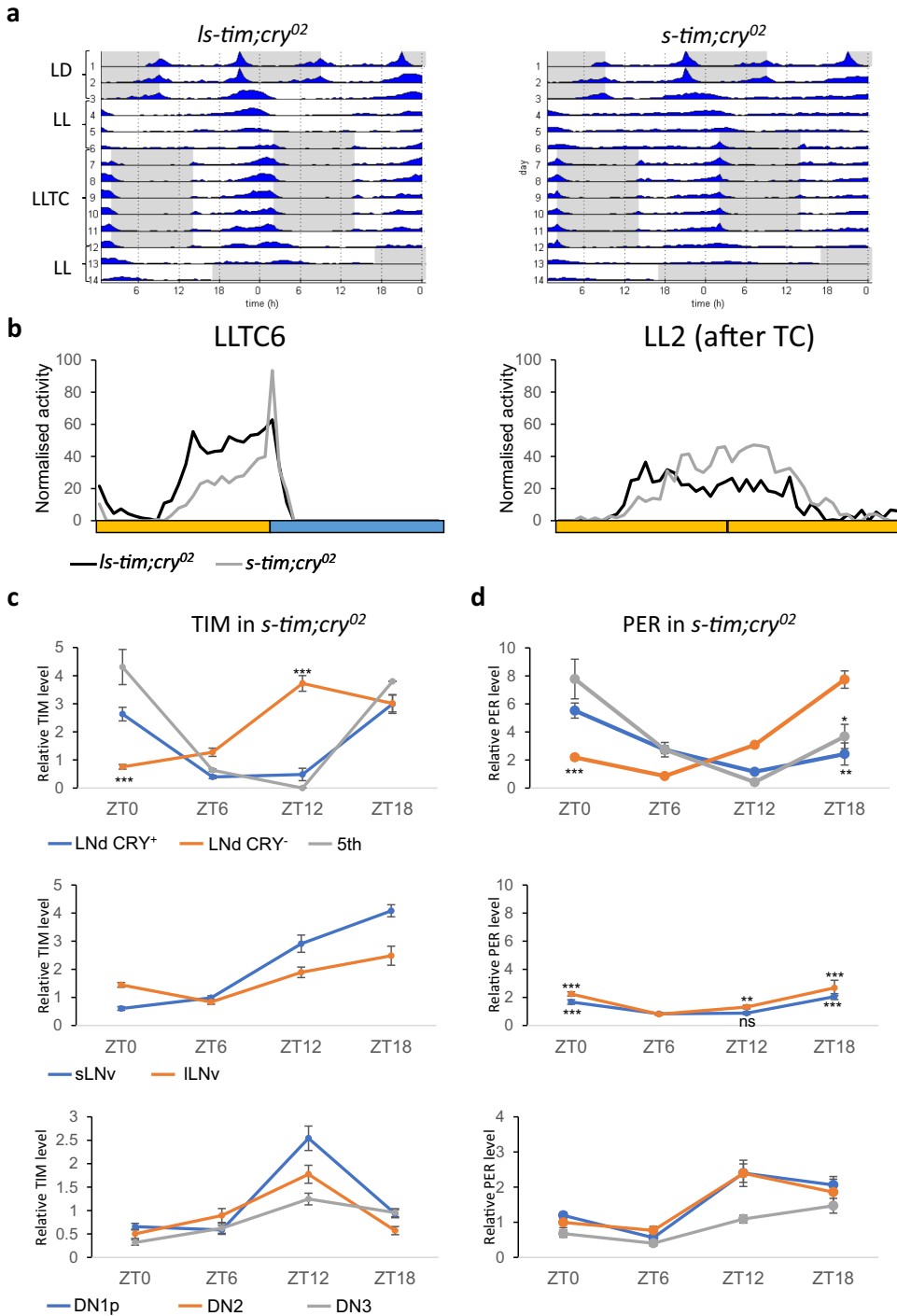

**Fig. 3 Cryptochrome depletion partially restores behavioural and molecular synchronisation during constant light and temperature cycles in *s-tim* flies.**
**a** Group actograms of one representative experiment as described in the legend for Fig. 1a. N (*ls-tim; cry^02*): 17, (*s-tim; cry^02*): 20. **b** Median of normalised activity of independent experiments combined during day 6 (left panel) of LLTC and day two of LL after LLTC (right panel). Yellow bar: thermophase, blue bar: cryophase (12 h each). N for left panel: (*ls-tim; cry^02*): 76; (*s-tim; cry^02*): 96. N for right panel: (*ls-tim; cry^02*): 36; (*s-tim;cry^02*): 54. Averages of normalised TIM (**c**) and PER (**d**) levels in the different groups of clock neurons during day six of LLTC in *s-tim; cry^02* flies. Number of brain hemispheres/time point = 3–5. Error bars = *sem* (standard error of the mean). A Mann–Whitney test was performed (with a Bonferroni correction) to compare the CRY-negative with the CRY-positive LNd and the 5th s-LNv at ZT0 and ZT12 (TIM) and ZT18 (PER). Because the PER level was lower in the l-LNv and the s-LNv compared to the other lateral neurons, we also performed a Mann–Whitney test to compare ZT6 with the other time points. * $p < 0.05$, ** $p < 0.01$, *** $p < 0.0001$. Source data are provided as a Source Data file.

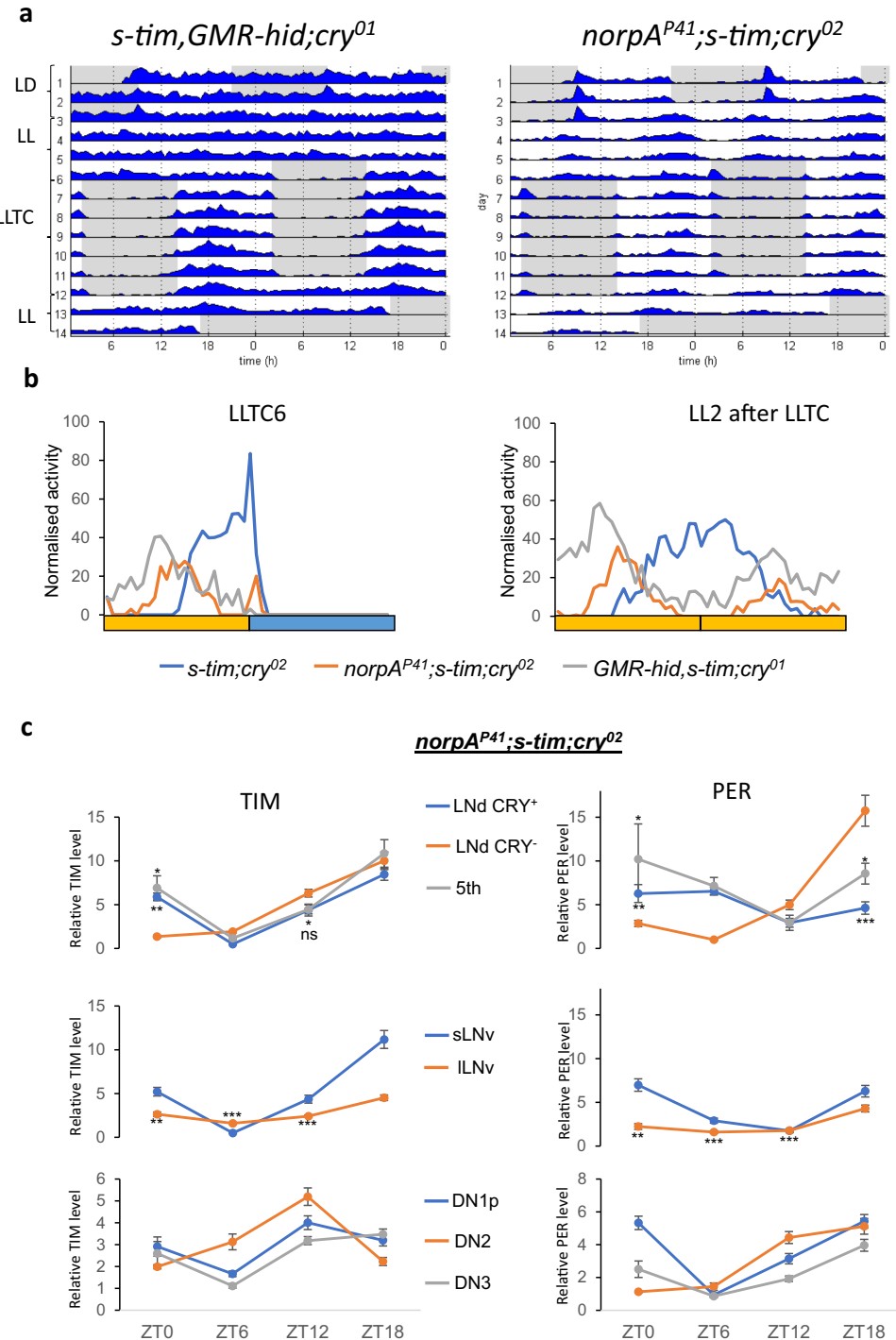

**Fig. 4 The visual system phases the locomotor output to the evening without affecting the molecular clock in *s-tim;cry0* background. a** Group actograms of one representative experiment as described in the legend for Fig. 1a. N (*s-tim, GMR-hid; cry$^{01}$*): 17, (*norpA$^{P41}$; s-tim; cry$^{02}$*): 19. **b** Median of normalised activity of independent experiments combined during day 6 (left panel) of LLTC and day two of LL after LLTC (right panel). Yellow bar: thermophase, blue bar: cryophase (12 h each). N for left/right panel (*s-tim, cry$^{02}$*): 56/45, (*norpA$^{P41}$; s-tim; cry$^{02}$*): 49/40, (*GMR-hid, s-tim; cry$^{01}$*): 32/30. **c** Averages of normalised TIM (left) and PER (right) levels in the different groups of clock neurons during day six of LLTC in *norpA$^{P41}$;s-tim;cry$^{02}$* flies. Number of brain hemispheres/time point = 2–5. Error bars = *sem* (standard error of the mean). A Mann–Whitney test was performed (with a Bonferroni correction) to compare the CRY-negative with the CRY-positive LNd and the 5th s-LNv at ZT0 and ZT12 (TIM), or ZT18 (PER). Note that at ZT12 the TIM level in the CRY-negative LNd is not significantly different from the 5th ($p = 0.08$). Because the PER level was lower in the l-LNv compared to the other lateral neurons, we also performed a Mann–Whitney test to compare ZT18 with the other time points. * $p < 0.05$, ** $p < 0.01$, *** $p < 0.0001$. Source data are provided as a Source Data file.

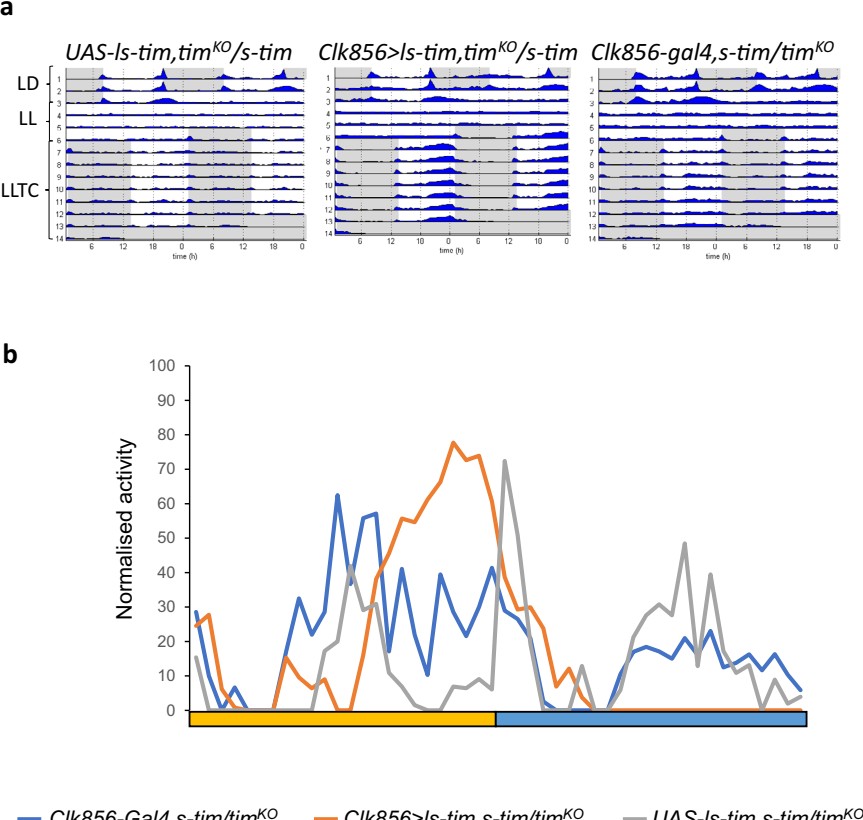

**Fig. 5 Expression of *ls-tim* within clock neurons is sufficient for synchronisation to temperature cycles in constant light. a** Group actograms of one representative experiment of the indicated genotypes as described in the legend for Fig. 1a. N (*UAS-ls-tim, tim^KO/s-tim*): 9, (*Clk856-Gal4; s-tim/UAS-ls-tim, tim^KO*): 20, (*Clk856-Gal4; s-tim/tim^KO*): 17. **b** Median of normalised activity during day 6 of LLT from the flies shown in (**a**). Yellow bar: thermophase, blue bar cryophase (12 h each). Source data are provided as a Source Data file.

neurons was sufficient to restore robust synchronization to LLTC, while expression in R1 to R6 had no effect (Fig. 5, S6a–c). While we cannot rule out a role for *ls-tim* in the R7 and R8 cells, the results unequivocally show that presence of the less-light-sensitive L-TIM form in clock neurons is sufficient for allowing temperature synchronisation in LL.

**The *ls-tim* allele enables flies to synchronise in white nights under semi natural conditions.** In order to determine if the *ls-tim* allele can be advantageous in natural conditions, we analysed behaviour under LLTC conditions experienced in the summer in Northern Europe. We decided to mimic the conditions of a typical summer day in Oulu, Finland (65° North) for two reasons. First, *Drosophila melanogaster* populate Northern Scandinavian regions in this latitude and overwinter here (e.g.,[37]). Second, from mid-May to the end of June day length varies from 19 to 22 h, and the rest of the "night" corresponds to civil twilight, where the sun does not set more than 6° below the horizon and general activities can be performed without artificial light ('white nights', maximum darkness between 1–3 lux). At the same time average temperatures vary by 10 °C between day and night, reaching an average maximum of 20 °C in July. To mimic these conditions, we programmed a 2.5 h period with 1 lux light intensity (civil twilight) and 12 h of 200 lux interspersed by ramps with linear increases (morning) or decreases (evening) of light intensity (Fig. 6a). Temperature cycled over 24 h with linear ramps between 12 and 19 °C, reaching its minimum towards the end of the civil twilight period and its maximum in the middle of the 200 lux phase (Fig. 6a). Using these conditions, we analysed two $w^+$

and one $w^-$ *s-tim* strains. Interestingly, all of these strains showed the same broad activity phase covering a large part of the 200 lux day period (Fig. 6b, c, S7). In addition, all *s-tim* strains showed a pronounced 2nd activity peak during the 3.5 h of down-ramping the light intensity from 200 lux to 1 lux (Fig. 6b, c, S7). In contrast, both *ls-tim* strains we tested ($w^+$ and $w^-$) showed only one defined activity peak during the 2nd half of the 200 lux phase and activity increase coincided precisely with the onset of the temperature decrease (Fig. 6b, c, S7). The results indicate that the synchronised circadian clock in *ls-tim* flies is responsible for a suppression of behavioural activity during the phase of increasing temperature. Interestingly, a similar repression of behavioural activity during ramped temperature cycles in DD depends on the gene *nocte*, which is required for temperature synchronisation during DD and LL[15,16]. *nocte* mutants steadily increase their activity with rising temperature[15], similar to what we observe here for *s-tim* flies (Fig. 6b, S7a), indicating a failure to synchronize their clock to the temperature cycle. To test this, we also analysed clock-less flies (*tim^KO*), which showed essentially the same behaviour as *s-tim* flies (Fig. 6b, c), indicating that *s-tim* flies are not able to synchronise their clock in Northern summer conditions as for example experienced in Oulu. Finally, to test if the same mechanism responsible for the lack of *s-tim* synchronization to rectangular laboratory LLTC conditions operates under semi-natural conditions, we also analysed *s-tim cry^02* flies under Oulu summer conditions. Strikingly, without CRY, *s-tim* flies showed essentially the same behaviour as *ls-tim* flies (Fig. 6b, c), indicating that the reduced light-sensitivity of the L-TIM:CRY interaction enables *ls-tim* flies to synchronise their clock in Northern summers.

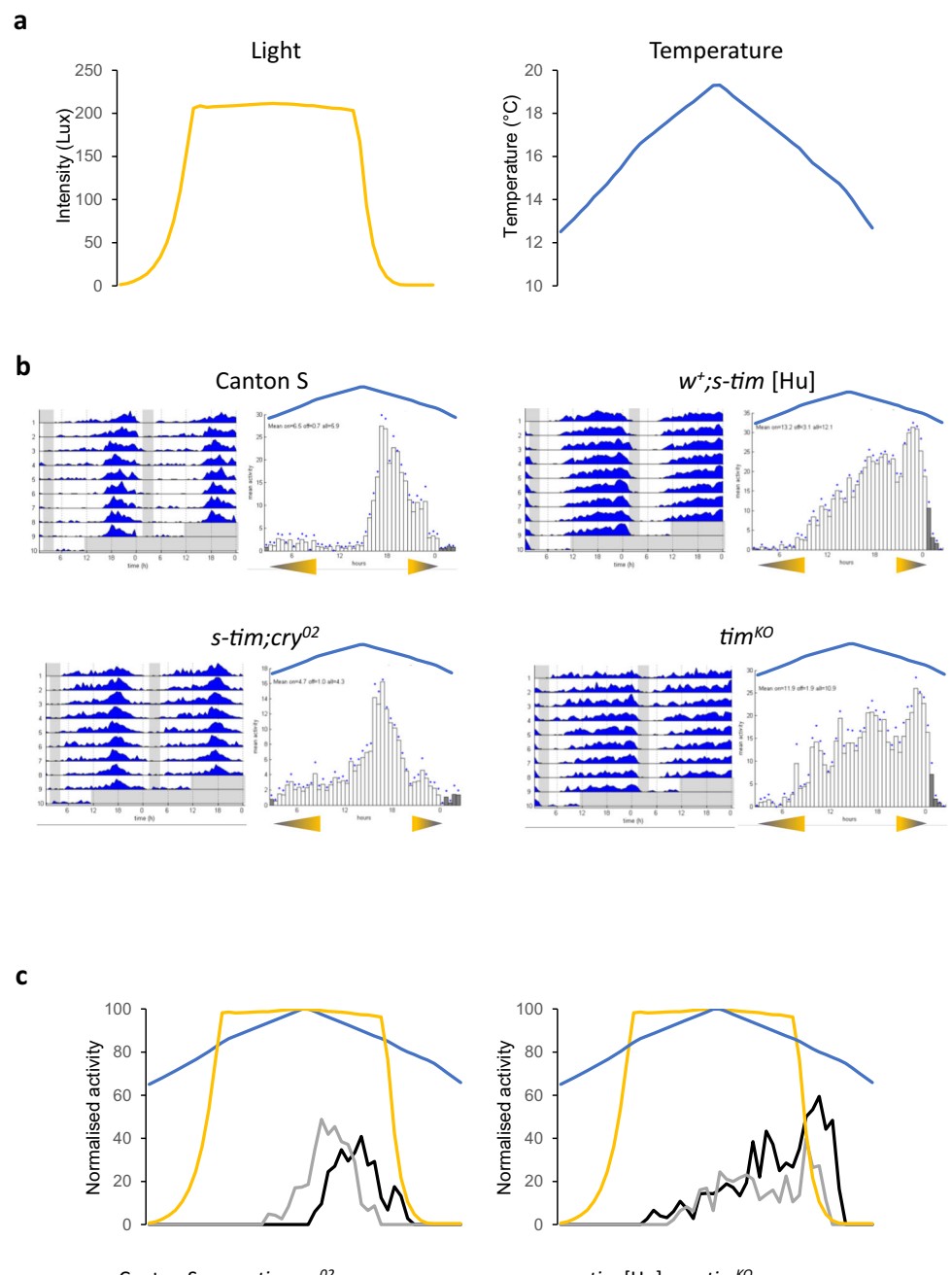

**Fig. 6 Only *ls-tim* flies are able to synchronise to Northern latitude summer conditions. a** Changes in daily light intensity (left graph) and temperature (right graph) recorded within the incubator programmed to reflect summer conditions in Oulu, Finland (65° North). **b** Group actograms and corresponding histograms of the last three days of one representative experiment in semi-natural Oulu conditions. White areas and bars: periods of light, and light ramping. Grey areas and bars, periods of 0.1 lux, reflecting civil twilight. Triangles under the histograms indicate the initiation of the light ramping (increase and decrease between 0.1 lux and 200 lux). The blue line above the histograms indicate the temperature ramping (between 12 and 19 °C) and the blue diamonds indicate SEM. N (Canton S): 19; (w+; s-tim [Hu]): 20; (s-tim; cry02): 21; (timKO): 14. **c** Median of normalised activity of independent experiments combined during the 6th day of Oulu condition. Yellow and blue lines indicate light and temperature ramping, respectively. N (Canton S): 38; (w+; s-tim [Hu]): 39; (s-tim; cry02): 41; (timKO): 33. Source data are provided as a Source Data file.

## Discussion

Light and temperature serve as two universal Zeitgebers to time the circadian clock in *Drosophila* and many other organisms. In *Drosophila*, exposure to constant light breaks down the clock machinery leading to arrhythmic locomotor activity[10,27]. This is likely due to constitutively low TIM levels in clock neurons caused by constant activation of the circadian photoreceptor CRY[5,8,9]. Cycling temperature, on the other hand, serves as another potent Zeitgeber to synchronise the circadian clock independent of light, suggesting the circadian thermo input is distinct from the light input at the circuit level. Interestingly, temperature cycles can "override" the effects of constant light and restore rhythmicity, both at the molecular and behavioural level[18,19]. Core clock proteins such as TIM and PER abolish their oscillation when exposed to constant light, but the rhythmic expression of these proteins is restored by temperature cycles in

both peripheral tissues and central pacemakers, suggesting the existence of a functional clock in these conditions. But the mechanisms that protect TIM from constant degradation by light during temperature cycles was so far an unresolved question.

We show here that only flies that carry the novel *ls-tim* allele can be synchronised to temperature cycles in LL, whereas flies carrying the ancient *s-tim* allele cannot. *ls-tim* is derived from *s-tim* by the insertion of single G nucleotide, which enables the usage of an additional upstream Methionine. As a result, *ls-tim* flies generate two TIM proteins: the original S-TIM (1398 amino acids) and L-TIM, carrying 23 additional N-terminal amino acids. In contrast, *s-tim* flies can only produce S-TIM[23]. L-TIM is less sensitive to light (more stable) compared to S-TIM, due to a weaker light-dependent interaction with the photoreceptor CRY[8,20]. This explains why *ls-tim* flies show reduced behavioural phase shifts in response to brief light pulses and why they are more prone to enter diapause in long photoperiods compared to *s-tim* flies[20,21]. The impaired L-TIM:CRY interaction is also the reason why *ls-tim* flies can synchronise to LLTC, because removal of CRY enables *s-tim* flies to synchronise as well (Fig. 3, S4). Nevertheless, both *s-tim* and *ls-tim* flies do become arrhythmic in LL at constant temperature, meaning that in *ls-tim* flies temperature cycles still somehow overcome the arrhythmia inducing effects of constant light. Presumably L-TIM levels in LL are below a threshold to support rhythmicity at constant temperatures, while above a threshold enabling the response to rhythmic temperature changes.

Our observation that removal of visual system function in the context of a *cry* mutant background leads to a behavioural phase advance, supports a role for visual system light input in phasing behaviour during temperature entrainment. Interestingly, in constant darkness and temperature cycles, wild type flies show the same early activity phase at the beginning of the thermo period as visual system impaired *cry* mutants in LLTC (Fig. 4a, b, S1c)[30]. Moreover, restricting clock function to the the 5th s-LNv, and the majority of the LNd and DN neurons in *cry* mutant flies, resulted in an activity peak late in the thermo phase, both in LLTC and DDTC conditions, similar to that of wild type flies in LLTC (Fig. 1a)[30]. The drastic phase difference in DDTC between wild type and *cry* mutant flies with a functional clock restricted to dorsal clock neurons, indicates that these dorsally located neurons (including at least some of the E-cells) are sufficient to drive behaviour in 16 °C: 25 °C temperature cycles, but that other clock neurons contribute to setting the behavioural activity phase in the absence of light or impaired light-input to the circadian clock neurons (Figs. 1, 3, 4)[30,38].

The *ls-tim* allele arose ~300–3000 years ago in southern Europe from where it is currently spreading northward by seasonal directional selection[21,22]. *ls-tim* enhances diapause, which presumably serves as driving force for this natural selection, by providing advantages in coping with the shorter day-length and earlier winter onset in higher latitudes[21,22]. In addition to an earlier onset of winter, northern latitudes close to the Arctic Circle are also characterized by extremely long photoperiods in the summer. Because it never gets completely dark in the summer, even regions more South, for example the Northern regions of the Netherlands, Germany and Poland, as well as the Baltic states experience considerable amounts of light at night. Our finding that *ls-tim* enables flies to synchronise to temperature cycles in constant light and particularly to semi-natural conditions mimicking white nights in Finland, indicates that this allele provides an additional fitness advantage during long photoperiods in central and Northern Europe. Considering the massive population expansion of *Drosophila* during the summer and that daily timing of activity offers a fitness advantage (e.g.,[39]), we propose that the ability to synchronise to temperature cycles in

long summer days constitutes a major positive selection drive for this allele. This positive drive is further boosted by the dominance of *ls-tim* over *s-tim*, i.e., heterozygous *ls-tim/s-tim* flies are able to synchronise as efficiently to LLTC as homozygous *ls-tim/ls-tim* flies do (Fig. 1).

Interestingly, other high-latitude *Drosophila* species also show reduced light sensitivity of their circadian clock, although via a different mechanism. These species (for example *D. ezoana* and *D. littoralis*) reduce light-sensitivity of the circadian clock by omitting CRY expression from the l-LNv clock neurons, thereby enabling their adaptation to long photoperiods[40,41]. Furthermore, several Northern latitude fly species have lost the ability to maintain free-running rhythms in constant darkness, implying that a circadian clock is not required in long summer day conditions[40–42]. It is not known however, if these species are able to synchronise to white nights or to temperature cycles in LL. Our results indicate that under Northern latitude summer conditions, the lack of a robust clock can be compensated by the ability to synchronise molecular behavioural and rhythms to temperature cycles. Nevertheless, it seems clear that independent strategies have evolved allowing insects to cope with light and temperature conditions in high-latitudes.

## Methods

**Fly stains**. Flies were reared on cornmeal-sucrose food at 18 °C or 25 °C under 12 h: 12 h LD cycles and 60% humidity until used in experiments. The following strains were used in this study: *norpA^P41* and *norpA^P41*, *cry^0234*, *cry^01*, *cry^02* and *w; iso s-tim[26]*, *Clk856-gal4[35]*, *y w; s-tim* and *y w; ls-tim[43]*, *w; iso31 ls-tim[25]*, *gmr-hid; cry^0144*, *Rh1-gal4* (BL8688). The *UAS-tim2.5* transgene encodes L-TIM and S-TIM[34] and is inserted on an *s-tim* chromosome. It was combined with *tim^KO28* using standard meiotic recombination. Wild type stocks used were Canton S (*ls-tim*, Jeffrey Hall lab), Tanzania (*s-tim*)[40], and Houten (Hu) (*s-tim* and *ls-tim* versions)[21]. If necessary *ls-tim* and *s-tim* chromosomes were exchanged using standard genetic crosses. *ls-tim/s-tim* genotypes were confirmed by PCR[23].

**Behavioural assays**. In total, 3–5 days old male flies were used for locomotor activity tests with the *Drosophila* Activity Monitor System (DAM,Trikinetics Inc). Fly activity was recorded in light- and temperature-controlled incubators (Percival,USA) every minute. Environmental protocols are indicated next to the first actogram in every figure. LLTC was phase delayed with respect to the original LD cycle by 5 h. Light intensity was between 400 and 800 lux (white fluorescence light). DDTC was advanced by 8 h with respect to the initial LD cycle. A 'flytoolbox' implemented in MATLAB (Math Works) was employed for plotting actograms and histograms[45]. Behaviour was quantified using a custom Excel macro[28]. In total, 30 min bin activity was normalized to the maximum level of activity for each fly. The median of this normalized activity was plotted, allowing to visualize the level of synchronization within a strain[28]. The same macro was used for plotting the light and temperature in Fig. (6, S7). To measure the slope in LLTC6, we manually determined the latest time point of the minimum median ($ZT_{min}$) and the first time point of the maximum median level ($ZT_{max}$) in minutes (for more clarity and to fit with the nomenclature we converted in the figures the ZTs into hours). We calculated for each fly the derivative of the line between $ZT_{max}$ and $ZT_{min}$: Slope = $(Act_{ZTmax}- Act_{ZTmin})/(t_{ZTmax}-t_{ZTmin})$. The box plots were made using Excel. The statistical tests were performed using open source Estimation Statistics (https://www.estimationstats.com/#/)[46].

**Immunohistochemistry and quantification**. Immunostaining experiments were performed as previously described[28]. Flies were placed in LL for 2 days and then entrained with a LLTC cycle and dissected on the 6th cycle. Brains were dissected in PBST 0.1% and fixed for 20 min at room temperature in PFA 4%. After 3 washes brains were blocked for 1 h at room temperature in PBST 0.1% + 5% goat serum. Primary antibodies were incubated for 48 h (in PBST 0.1% + 5% goat serum) at 4 °C, while secondary incubation was done overnight at 4 °C. Brains were mounted using Vectashield. Rat anti-TIM generated against TIM-fragment 222-577[47] (kind gift of Isaac Edery) was used at 1/2000. Monoclonal anti-PDF (DSHB) was used at 1/1000, and pre-absorbed Rabbit anti-PER[48] was used at 1/15000. Secondary antibodies used: goat anti-mouse 488 1/2000 (Invitrogen, cat # A-32723), goat anti-rabbit 555 1/2000 (Invitrogen, cat # A-21428) and anti-rat 647 1/1000 (Invitrogen, cat # A-21247). Brains were imaged with a Leica TCS SP8 confocal microscope with a 63x objective. Average intensity was measured using ImageJ and quantification was normalized to the background: (signal-background)/background[49].

**Statistical analysis**. All statistical tests were performed using open source Estimation Statistics (https://www.estimationstats.com/#/)[46]. For comparison between

two genotypes the 'Two Groups' (counterpart to Student's *t* test) and for comparison of staining intensities between multiple time points the 'Shared Control' (Cumming Plot) function was used[46].

**Reporting summary**. Further information on research design is available in the Nature Research Reporting Summary linked to this article.

## Data availability

Source data used to generate all plots and graphs are provided with this paper. The raw datasets generated consist of raw behavioural activity files and confocal microscope images. Due to the large size of these files they were not deposited in a public repository, but are available from the corresponding authors on reasonable request. The data are saved on a SSD disk, on a computer and on the laboratory network. Source data are provided with this paper.

## Code availability

Behaviour was analysed with the 'flytoolbox' implemented in Matlab as described in[45]. Quantification of anticipatory behaviour (slope) was performed using a custom Excel Macro as described[28]. Flytoolbox and custom Excel Macro have been deposited in GitLab.

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

## Acknowledgements

We thank Isaac Edery for anti-TIM antibodies and Charlotte Helfrich-Förster, Charalambos Kyriacou, and Francois Rouyer for fly stocks. We thank Charlotte Helfrich-Förster and Peter Deppisch for sharing unpublished results. This work was supported by a BBSRC research grant (BB/H001204/1) and by the European Union's Horizon 2020 Research and Innovation Programme under the Marie Skłodowska-Curie grant agreement no. 765937 (CINCHRON) given to R.S. This work was partly funded by the German Research Foundation (DFG) as part of the CRC TRR 212 (NC[3]) – project number 316099922 given to R.S.

## Author contributions

Investigation: A.L., C.C., S.L., M.X., R.G., and R.S.; Formal analysis: A.L., C.C., S.L., and R.S., Visualization and writing: A.L. and R.S.; Resources, funding acquisition: R.S.; Conceptualization and design of the study. A.L., C.C., and R.S.

## Funding

## Competing interests

The authors declare no competing interests.
