## [Peer Review File · Nature Communications]

Reviewers' Comments:

Reviewer #1:

Remarks to the Author:

It has been shown a long time ago by the Tomioka lab that temperature cycles prevent the degradation of the Timeless clock protein and subsequent behavioral arrhythmicity in constant light conditions in *Drosophila*. This indicated the existence of intriguing interactions between Cryptochrome-dependent light entrainment and temperature entrainment of the brain circadian clock. The present study brings new and interesting data showing that the phototransduction pathway protein NORPA is required for this inhibition of the light-driven arrhythmicity by temperature cycles. The authors show that NORPA mediates the temperature cycling effect through its unexpected expression in the clock neurons, by inducing a decrease of CRY levels, which results in increased TIM levels.

These are very interesting results revealing an intriguing function of the NORPA pathway for mediating temperature effects. I nevertheless believe that some of the data need to be reinforced both in the behavioral analysis and the molecular experiments, as detailed below.

1- In Fig 1B-C, the *norpA/UASnorpA* controls seem to show a kind of advanced peak during the warm phase rather than no peak as *norpA tim* do. Could this reflect a leak of expression of the *UASnorpA*?

2- The calculation of the EI is unclear to me. The authors indicate that they use the time window of the peak in the *yw* flies (ZT6-11) as a reference to define the time window of the peak in the three experimental genotypes. If I understand properly, a peak that would not be located in this time window would be missed with this method. It seems to me that the peak time window should be defined without any assumption on its position, otherwise a phase difference would be interpreted as an absence of peak. The results are indeed surprising with the EI of the *norpA tim67* flies (blue) being higher than the one of the *norpA/UASnorpA* flies (orange). This is not easy to understand when looking at the daily average plots in Fig1.

Sorry if missed something but I believe that the EI should be calculated differently.

3- Also in Fig1: to my knowledge, *timgal4* expression is far from being restricted to clock cells.

4- Fig 2. The TIM staining result in LL 25-16°C showing a peak in protein levels at the end of the thermophase in wild type flies is very surprising. As indicated by the authors, other studies of temperature cycling show PER/TIM peaks at the end of the cryophase (comparable to LD cycles where the peak is end of the night). In particular, Yoshii et al 2005 described PER peaking at ZT20-2 in all clock neurons of flies in LL 30-25°C, with a behavioral peak at the end of the thermophase as observed in the present work. This of course weakens the *norpA* results. The authors need to confirm this unexpected clock protein cycling result in these *stim yw* flies by doing a PER staining experiment with the same conditions, at least for the main neuronal subsets. I would also suggest to add the *Istim yw* genotype.

5- Fig2. TIM does not cycle in most clock neurons in *norpA* mutants. The authors conclude that *norpA* is required for synchronization in these LLTC conditions. Could it be that *norpA* is required for TIM cycling in an individual cell? In other words, is it possible to see whether the non-cycling that is observed at the neuronal group scale is a consequence of non-cycling in all individual cells (clock defect in these conditions) within a group or non-synchronized cycling in individual cells (sync defect)?

6- Fig4. The *ninaE17* behavioral result appears rather ambiguous and I believe that *Rh1* function is an important point. I think that the authors should reinforce the data to be able to conclude about the non-involvement of *Rh1* in LLTC entrainment. Using another allele and/or showing that PER/TIM cycling still occurs in the mutant would allow to draw a stronger conclusion.

7- Fig8. Nice correlation between high CRY levels and low TIM levels in *norpA* mutants. The behavioral rescue through *gal4*-driven *norpA* expression (Fig 5) and the single cell RNAseq data (Fig 6) support *norpA* function in the clock neurons. However, to show that NORPA directly induces a decrease in CRY levels and an increase in TIM levels, the authors should assay CRY and TIM in the clock neurons of flies with *norpA* being rescued in the same cells. A *gal4* very specifically expressed in clock neurons (*clk856*, pdf) would be best suited for this experiment

Minor points:

The *s/l*s *tim* genotypes should be briefly presented in the introduction

Color codes in the actograms of fig S1 are not easy to follow. Ideally, white color should be replaced by yellow for lights-ON 25°C and grey by yellow+grey for lights-OFF 25°C (if I understand correctly).

p10 "These data indicate that TIM is the target of PLC-β and CRY-mediated degradation during LL and TC. "

It is not clear to me why the data at this point of the paper say something about CRY-mediated degradation. Please clarify.

Reviewer #2:

Remarks to the Author:

The paper of Chen et al. answers the following longstanding question in circadian biology: How can a fly remain rhythmic under constant light, a condition that occurs in nature at high latitudes? Fruit flies have a very light-sensitive circadian clock that stops working under constant light, because light leads to the degradation of the circadian clock protein Timeless (TIM). This is accomplished by the photoreceptor Cryptochrome (CRY). Light-activated CRY binds to TIM and the F-box protein JETLAG (JET) triggering the degradation of TIM and CRY in the proteasome. Consequently, the clock stops ticking. However, paradoxically, flies remain rhythmic under constant light conditions as long as daily temperature cycles are present. So far, it was completely unknown, how temperature cycles prevent CRY from degrading TIM.

Here, the authors show that the Gq protein-dependent signaling pathway that activates the enzyme Phospholipase C-β (PLC-β) operates not only in the eyes but also in the central clock neurons. There, it represses CRY in the presence of light and temperature cycles and allows the flies to remain rhythmic. This finding is brand-new and demonstrates how flies can maintain clock function and rhythmic behavior at northern latitudes. Thus, the study closes a gap in the understanding of the fly circadian clock that is also relevant for other species. Furthermore, the study is of high technical standard well conducted and statistically properly analyzed.

Nevertheless, the paper is not easy to understand. I cannot detect a mistake in the logic of the conclusions, but the following puzzles me: CRY is not detectable in arrhythmic wildtype flies under constant light and it is similarly undetectable in rhythmic wildtype flies under constant light combined with temperature cycles. In contrast, CRY remains visible in arrhythmic *norpA* mutants under constant light combined with temperature cycles. I understand that in wildtype flies, light-activated CRY provokes the degradation of TIM and subsequently CRY itself is degraded. I also understand that the additional presence of temperature cycles activates PLC-β (=Norpa), which then represses CRY. The total absence of CRY leads to rhythmic behavior in constant light, because TIM is not anymore degraded. Furthermore, in the absence of Norpa and the presence of temperature cycles, there is no repression of CRY. Consequently, CRY leads to degradation of TIM and to arrhythmic behavior as it does in wildtype flies. However, I do not understand why CRY remains visible and is not degraded after its light activation. What is different from wildtype flies? I believe that the paper would benefit from a cartoon illustrating the action of Norpa and CRY under the different conditions.

Minor comments:

Line 34: Gq-PLC- β needs to be explained.

Line 43: something is missing in this sentence (perhaps: circadian clocks, which regulate)

Line 46: 'terminate' instead of 'terminateMi'

Line 52: I think that it is oversimplified to talk about CRY as the main circadian photoreceptor. This is even in contrast to the results of the corresponding author.

Line 72: why not adding 'operating in the eyes' after 'cascade'. Although, the authors demonstrate later that PLC- β operates not only in the eyes, this would help understanding here.

Line 74: ')' is missing after '18'.

Line 88: explain 'Gq'

Line 132 and Figure 1B: The entrainment of norpA41 mutants with norpA rescued with tim-gal4 looks even better than that of wildtype flies. Is this because you overexpress norpA in the clock neurons, in which it is possibly expressed at low levels? Or, is this due to norpA expression in the eyes?

Lines 176: The sentence 'Introducing one copy of Is-tim is not sufficient...' disturbs the reading flow. I propose to move it. Just one sentence later, it appears adequate.

Line 179: Do not abbreviate 'EI' in the text. See also line 239.

Line 235: 'Gentile et al., 2013b' needs to be cited as a number.

Line 249: Please explain 'rh1-4'

Line 252: '(add references)' need to be exchanged by real references.

Line 252: 'In contrast' does not fit. I would rather write 'Similarly'

Line 261: 'rescues' appears inappropriate. What about 'restores'?

Line 278: What about 'enabling' instead of 'regulating'?

The References need a thorough editing. In many places, the spaces between the words are missing. The abbreviated Journal names have sometimes a full stop after the abbreviations, sometimes not. Reference 34 lacks the Journal name and something is wrong with the author names. Species names are not italicized. In reference 44, *Drosophila* is written in lower case letters...

Reviewer #3:

Remarks to the Author:

This is a very interesting ms from the Stanewsky lab which focuses on a long neglected but intriguing feature of fly chronobiology, namely how temperature cycles can overcome the arrhythmic behaviour induced by constant light (LL0). The ms is crisply written and the narrative flows very nicely, indeed I got quite excited wondering what the next section would bring. The ms focuses on the effects of norpA on temperature entrainment and how this interacts with CRY and TIM. The experiments also show the fundamental differences between the Is-TIM and s-TIM isoforms in temperature entrainment, which have so far been limited to circadian and photoperiodic light sensitivity. There is also an additional evolutionary context at the end of the Discussion, that will appeal to both neurogeneticists as well as behavioural biologists interested in evolution/population genetics. What the ms does not show is the molecular mechanism by which norpA performs its inhibitory role on CRY. Below are my comments

Fig S1. Please add arrows to the startle and the main activity component for the mutants as the differences in the mutants may not be obvious to a non specialist. The norpA main locomotor phenotype is advanced in LD, which is probably due to enhanced per splicing (refs 26,27). Might the larger advance of the mutant main activity in TC be due to even greater per splicing under these conditions which would carry over to the first days of DD? Refs 16,28 suggest no daily changes in per splicing under TC but my question is different – is the overall level of per splicing (irrespective of any daily changes) enhanced under TC where the average daily temperature is ~20°C?

L121 severe

L127 Fig 1A -the p24 mutant looks much more like WT than the p41 mutant

Fig 2B the DN2s in yw do not show an oscillation, which is odd? Was the interaction term in the 2-way ANOVA significant? Could the asterisks be printed in a different colour to red and black as they are difficult to distinguish from the black datapoints?

Fig 3A It's interesting that the mutants on Is-tim show a reduced startle to cold-hot, but increased

startle to hot-cold transition...more sensitive to cold? reminds me of the splicing of per with norpA which means that the mutant behaves as if it is colder than it really is.

L219-220 and Fig 4. Perhaps a little more guarded about ninaE mutants given the behavioural phenotypes are not as clear cut as in wild-type.

L227 Abruzzi et al PLoS Genet (2017) 13:e1006613 have shown norpA to be cycling in LNds so it is expressed there as well. (You need to dig into the supplementaries of that paper to find it). Also 'recently' is inappropriate for ref 42 which is 10 years old!

L235 Gentile et al reference needs to be numbered 31

L252 'add references'?

L245-259 this could be compressed and figure 6 placed in supplementaries as it is a database RNAseq study and, although relevant, is the least interesting of the main results.

Fig 7C comment required on why TIM peaks in the cold compared to the warm in the norpA; cry mutants which restores a DDTC like profile?

Fig 8E comment required on why the double mutant shows less CRY under LD 25oC

L336 Is-tim allele arose approx. 10,000 years ago. This has been amended more recently by Zonato et al (2018) J Biol Rhythms 33(1):15-23. The allele may have arisen, I believe between 300-3000 years ago

L341 D. melanogaster to my knowledge have not been found north of the Arctic circle but are found in southern to mid Scandinavia where summer photoperiods will be extremely long. If under these longer photoperiods the clock becomes less robust, but temperature cycles can restore some rhythmicity, this may be advantageous. However Helfrich-Forster has argued in a couple of recent papers that perhaps other more northern species of Drosophilids do not need to be rhythmic under these conditions. Perhaps a nod to this alternative view might be warranted. My personal feeling is that tropically evolved flies (like D. melanogaster) need to be rhythmic and will seek to maintain rhythmicity under extreme photoperiods, so the mechanism proposed here will help.

A line or two about how norpA might mechanistically block CRY degradation would be welcomed.

Response to Reviewers:

We thank all three reviewers for their interest in our study and for the very constructive comments and suggestions in response to our original submission. We also apologize for the exceptionally long time it took us to submit the revised version of our manuscript. The reasons for this delay are twofold: first, due to Covid-19 inflicted temporary shutdown of our lab, as well as prolonged periods of only temporary lab access, experiments took much longer than expected. Second, the experiments suggested by the reviewers lead to a rather drastic changes of the content as you will notice upon reading the revised manuscript.

The major changes are:

- Using different methods (outlined in detail in response to point 7 of referee 1 below) we were not able to confirm a role for *norpA* in destabilizing CRY within clock neurons. Because our original and revision experiments reveal only minor differences between *norpA*; *s-tim* mutants and wild type *s-tim* flies with regard to molecular and behavioral synchronization to LLTC, we decided to focus the new version of the manuscript on the striking differences between the natural *s-tim* and *ls-tim* polymorphisms in the clock gene *timeless*.
- Careful comparison of TIM and PER expression within clock neurons of *s-tim* and *ls-tim* flies revealed the absence of a functional molecular clock in *s-tim* flies under constant light and temperature cycles (LLTC). This could be confirmed on a behavioral level, by essentially identical behavior of *s-tim* and *tim* loss-of-function mutants in LLTC.
- Removal of CRY largely restores *s-tim* molecular and behavioral synchronization to LLTC, implicating that the reduced interaction of L-TIM with CRY as the underlying molecular mechanism for the ability of *ls-tim* fly to synchronize to LLTC. We show that *ls-tim* expression restricted to the clock neurons only, is sufficient for robust synchronization to LLTC.
- *ls-tim*, but not *s-tim* flies are able to synchronize their behavior to semi-natural conditions mimicking so called 'white nights' summer conditions (3-4 hours of civil twilight, rest of the day LL, temperature cycle with 10 °C amplitude) in Scandinavia. We therefore propose that the ability to synchronize to LLTC is the major driving force for the ongoing seasonal directional selection of *ls-tim* towards Northern latitudes.

Again, we thank the reviewers for their suggestions, which lead to the experiments resulting in this substantially revised version of the manuscript. While the underlying molecular mechanism is different, our work explains how flies can synchronize to temperature cycles in constant light on a molecular level. Furthermore, a new set of experiments mimicking the summer conditions of Oulu in Finland, strongly implicate this mechanism as a driving force for the directional selection of the *ls-tim* allele.

Although the manuscript has changed substantially, we still decided to make a point-by-point response to each of the referee's comments (see below).

Reviewer comments and point-by-point responses

Reviewer #1 (Remarks to the Author):

It has been shown a long time ago by the Tomioka lab that temperature cycles prevent the degradation of the Timeless clock protein and subsequent behavioral arrhythmicity in constant light conditions in *Drosophila*. This indicated the existence of intriguing interactions between Cryptochrome-dependent light entrainment and temperature entrainment of the brain circadian clock. The present study brings new and interesting data showing that the phototransduction pathway protein NORPA is required for this inhibition of the light-driven arrhythmicity by temperature cycles. The authors show that NORPA mediates the temperature cycling effect through its unexpected expression in the clock neurons, by inducing a decrease of CRY levels, which results in increased TIM levels.

These are very interesting results revealing an intriguing function of the NORPA pathway for mediating temperature effects. I nevertheless believe that some of the data need to be reinforced both in the behavioral analysis and the molecular experiments, as detailed below.

*We thank this referee for acknowledging the importance of our work. Even though the mechanism of how *Drosophila melanogaster* can maintain rhythmic behavior and clock function in Northern latitudes is different from the one we originally proposed, we think it is equally important. In addition, the requirement of *Is-tim* for synchronizing to long summer days typical for Northern latitudes, supplies an additional (or alternative) reason for the directed selection of this novel *tim* allele.*

1- In Fig 1B-C, the *norpA*/UAS*norpA* controls seem to show a kind of advanced peak during the warm phase rather than no peak as *norpA tim* do. Could this reflect a leak of expression of the UAS*norpA*? Yes, we agree that the *norpA*; UAS-*norpA* flies show an advanced peak, which presumably is due to leaky UAS-*norpA* expression. Since we now focus the manuscript on the more compelling difference between *Is-tim* and *s-tim*, the *norpA* rescue experiments are no longer in the manuscript. We do see a clear influence of *norpA* in the background of *cry* mutants (new Figure 4) and also in red light and temperature cycles (RRTC, see Figure below). We plan to repeat the UAS-*norpA* rescue experiments in RRTC in a follow up study of this work.

Figure: behavior of *s-tim* and *norpA*^{P41}; *s-tim* flies in red-light and temperature cycles (RRTC). In contrast to the subtle differences between *s-tim* and *norpA*^{P41}; *s-tim* flies in LLTC (see Figures 1, S1, S2, and S5 of the revised manuscript) the differences are more pronounced in RRTC, which eliminates the function of CRY, but not that of Rhodopsin 1 (Rh1) and Rhodopsin 6 (Rh6). The results implicate a function for *norpA* and potentially also Rh1 and Rh6 in synchronization to LLTC. This role and the role

of the visual system for LLTC synchronization in general (cf. Figure 4 of the revised manuscript), will be subject of an independent future study.

2- The calculation of the EI is unclear to me. The authors indicate that they use the time window of the peak in the *yw* flies (ZT6-11) as a reference to define the time window of the peak in the three experimental genotypes. If I understand properly, a peak that would not be located in this time window would be missed with this method. It seems to me that the peak time window should be defined without any assumption on its position, otherwise a phase difference would be interpreted as an absence of peak. The results are indeed surprising with the EI of the *norpA tim67* flies (blue) being higher than the one of the *norpA/UASnorpA* flies (orange). This is not easy to understand when looking at the daily average plots in Fig1.

Sorry if I missed something but I believe that the EI should be calculated differently.

We do not use EI calculation to quantify synchronized behaviour in the revised version of the manuscript

3- Also in Fig1: to my knowledge, *timgal4* expression is far from being restricted to clock cells.

The reviewer is correct, tim-gal4 is expressed in central and peripheral clock cells, not just in central brain clock neurons (Kaneko and Hall, 2000). We do not use this transgene in the revised version of the manuscript.

4- Fig 2. The TIM staining result in LL 25-16°C showing a peak in protein levels at the end of the thermophase in wild type flies is very surprising. As indicated by the authors, other studies of temperature cycling show PER/TIM peaks at the end of the cryophase (comparable to LD cycles where the peak is end of the night). In particular, Yoshii et al 2005 described PER peaking at ZT20-2 in all clock neurons of flies in LL 30-25°C, with a behavioral peak at the end of the thermophase as observed in the present work. This of course weakens the *norpA* results. The authors need to confirm this unexpected clock protein cycling result in these *stim yw* flies by doing a PER staining experiment with the same conditions, at least for the main neuronal subsets. I would also suggest to add the *lstim yw* genotype.

*We thank the reviewer for this suggestion. Doing these additional staining experiments revealed that the main difference in the time courses were actually between *lstim* and *sstim* and not between *norpA* mutant and wild type flies. Following the reviewers suggestion when we performed the stainings for *lstim* with anti-PER, we got very similar results to those cited by the referee (Yoshii et al 2005), namely peaks of PER late in the cryophase, or early in the thermophase. Interestingly, peaks for TIM in *lstim* flies were phase advanced compared to those of PER and also compared to those for TIM in LD, peaking at the beginning of the cryophase. The real surprise came when we repeated the anti-TIM and anti PER stainings on *sstim* flies. PER was at extremely low levels and basically not cycling at all (except for the DN3). For TIM the weak, but early peak at ZT6 was nicely reproducible, but closer inspection revealed that TIM was cytoplasmic at all times in *sstim* flies during LLTC. Combined with the non-cycling of PER in *sstim* flies, we concluded that the molecular clock is not running in *sstim* flies in LLTC and we therefore decided to focus on the difference between *sstim* and *lstim*. We still believe that *norpA* influences TIM cycling in LLTC, because the low amplitude oscillation with a (cytoplasmic) peak at ZT6 observed in *sstim* flies (original and new Figure 2A), is absent in *norpA; sstim* flies (original Figure 2A, new Figure 4C, same data). In addition, in *norpA; cry* double mutants the peak of TIM expression is advanced compared to cry single mutants (new Figure*

3C compared to new Figure 4C). The exact role of *norpA* in LLTC entrainment will be subject of a follow up study (see above).

5- Fig2. TIM does not cycle in most clock neurons in *norpA* mutants. The authors conclude that *norpA* is required for synchronization in these LLTC conditions. Could it be that *norpA* is required for TIM cycling in an individual cell? In other words, is it possible to see whether the non-cycling that is observed at the neuronal group scale is a consequence of non-cycling in all individual cells (clock defect in these conditions) within a group or non-synchronized cycling in individual cells (sync defect)?

*See response to point 4 above. The comparison of PER and TIM cycling between *ls-tim* and *s-tim* revealed the lack of a functional clock in *s-tim* flies during LLTC. Due to the continuous cytoplasmic localization of TIM and the similar behaviour of *s-tim* and *timKO* flies (Figure 1, S1, S2) we are convinced that *s-tim* flies have a clock defect in LLTC conditions*

6- Fig4. The *ninaE17* behavioral result appears rather ambiguous and I believe that *Rh1* function is an important point. I think that the authors should reinforce the data to be able to conclude about the non-involvement of *Rh1* in LLTC entrainment. Using another allele and/or showing that PER/TIM cycling still occurs in the mutant would allow to draw a stronger conclusion.

*We agree with the reviewer that *Rh1* function is an important point with regard to LLTC behaviour. We have addressed this in the revised version with two different experiments. First (Figure 4A), we ablated all *Rh1* expressing photoreceptor cells (plus those expressing *Rh2-Rh6*) using *gmr-hid* in a *cry* mutant background. Like *norpA*; *cry* double mutants, the photoreceptor ablated flies show an advanced evening peak, pointing to a (*norpA*-dependent) role of the visual system (including *Rh1* cells) in setting the behavioural phase during LLTC entrainment. Second (Figure S6), we expressed *ls-tim* specifically in *Rh1* cells (*Rh1-gal4*) or clock neurons (*Clk856-gal4*) and could show that *ls-tim* is required in clock neurons but not in *Rh1* cells for LLTC entrainment to occur.*

7- Fig8. Nice correlation between high CRY levels and low TIM levels in *norpA* mutants. The behavioral rescue through *gal4*-driven *norpA* expression (Fig 5) and the single cell RNAseq data (Fig 6) support *norpA* function in the clock neurons. However, to show that NORPA directly induces a decrease in CRY levels and an increase in TIM levels, the authors should assay CRY and TIM in the clock neurons of flies with *norpA* being rescued in the same cells. A *gal4* very specifically expressed in clock neurons (*clk856*, pdf) would be best suited for this experiment.

*We thank the reviewer for these suggestions to further check if NORPA stabilizes CRY in clock neurons during LLTC. We did perform the experiment suggested by the referee, i.e. check CRY levels in *norpA*, *Clk856>UAS-norpA* compared to single *norpA* mutants. Unfortunately we could not repeat the initial observation that lack of NORPA stabilized CRY. We had to ask the Helfrich-Förster lab for a new batch of anti-CRY antibody, and suspected that the new batch was not working as well as the original one we used. To circumvent this problem we first used a UAS-luciferase-Cry fusion that has previously been used to study the stability of CRY (Peschel et al 2009: doi: 10.1016/j.cub.2008.12.042). But expression of this construct using the *Clk856* driver revealed no difference between wild type and *norpA* mutants during LLTC in an *s-tim* background. In a 3rd attempt, we used a GFP-CRY fusion generated in the lab of Paul Hardin, which fully rescues CRY function (Agrawal et al 2016: <http://dx.doi.org/10.1016/j.cub.2017.06.064>). Again, we did not observe a difference in GFP-CRY levels within clock neurons during LLTC in an *s-tim* background between wild type flies and *norpA**

*mutants. We therefore conclude that the destabilizing effect of NORPA function on CRY stability during LLTC is not reproducible. We consequently removed the original Figure from the revised manuscript, which now focusses on the stabilizing effect of *Is-tim* instead.*

Minor points:

The *s/Is tim* genotypes should be briefly presented in the introduction

*The differences between *Is-tim* and *s-tim* are now being thoroughly described, because they became the main focus of the revised manuscript*

Color codes in the actograms of fig S1 are not easy to follow. Ideally, white color should be replaced by yellow for lights-ON 25°C and grey by yellow+grey for lights-OFF 25°C (if I understand correctly).

Figure S1 has been removed from the revised manuscript. In the revised manuscript the environmental conditions are not indicated to the left of the actograms

p10 “These data indicate that TIM is the target of PLC-β and CRY-mediated degradation during LL and TC. “

It is not clear to me why the data at this point of the paper say something about CRY-mediated degradation. Please clarify.

That TIM is a target of PLC-β mediated degradation during LLTC is no longer part of the revised manuscript. In fact our attempts to reproduce the original finding failed (see response to point 7, above).

Reviewer #2 (Remarks to the Author):

The paper of Chen et al. answers the following longstanding question in circadian biology: How can a fly remain rhythmic under constant light, a condition that occurs in nature at high latitudes? Fruit flies have a very light-sensitive circadian clock that stops working under constant light, because light leads to the degradation of the circadian clock protein Timeless (TIM). This is accomplished by the photoreceptor Cryptochrome (CRY). Light-activated CRY binds to TIM and the F-box protein JETLAG (JET) triggering the degradation of TIM and CRY in the proteasome. Consequently, the clock stops ticking. However, paradoxically, flies remain rhythmic under constant light conditions as long as daily temperature cycles are present. So far, it was completely unknown, how temperature cycles prevent CRY from degrading TIM.

Here, the authors show that the Gq protein-dependent signaling pathway that activates the enzyme Phospholipase C-β (PLC-β) operates not only in the eyes but also in the central clock neurons. There, it represses CRY in the presence of light and temperature cycles and allows the flies to remain rhythmic. This finding is brand-new and demonstrates how flies can maintain clock function and rhythmic behavior at northern latitudes. Thus, the study closes a gap in the understanding of the fly circadian clock that is also relevant for other species. Furthermore, the study is of high technical standard well conducted and statistically properly analyzed.

*We thank this referee for acknowledging the importance of our work. Even though the mechanism of how *Drosophila melanogaster* can maintain rhythmic behavior and clock function in Northern latitudes is different from the one we originally proposed, we think it is equally important. In addition,*

*the requirement of *Is-tim* for synchronizing to long summer days typical for Northern latitudes, supplies an additional (or alternative) reason for the directed selection of this novel *tim* allele.*

Nevertheless, the paper is not easy to understand. I cannot detect a mistake in the logic of the conclusions, but the following puzzles me: CRY is not detectable in arrhythmic wildtype flies under constant light and it is similarly undetectable in rhythmic wildtype flies under constant light combined with temperature cycles. In contrast, CRY remains visible in arrhythmic *norpA* mutants under constant light combined with temperature cycles. I understand that in wildtype flies, light-activated CRY provokes the degradation of TIM and subsequently CRY itself is degraded. I also understand that the additional presence of temperature cycles activates PLC- β (=NorpA), which then represses CRY. The total absence of CRY leads to rhythmic behavior in constant light, because TIM is not anymore degraded. Furthermore, in the absence of NorpA and the presence of temperature cycles, there is no repression of CRY. Consequently, CRY leads to degradation of TIM and to arrhythmic behavior as it does

in wildtype flies. However, I do not understand why CRY remains visible and is not degraded after its light activation. What is different from wildtype flies?

I believe that the paper would benefit from a cartoon illustrating the action of NorpA and CRY under the different conditions.

*The reviewer is correct, that looking at the CRY levels in LLTC in wild type versus *norpA* mutants is puzzling. What we thought originally was that the increased CRY levels in *norpA* mutants during LLTC, are a consequence of the enhanced binding of TIM to the F-box protein JET (responsible for TIM and CRY degradation in LL; Peschel et al 2009: doi: 10.1016/j.cub.2008.12.042). According to the model developed in this study, JET binds preferentially to TIM, and after TIM is degraded, JET binds to CRY to initiate CRY degradation. We thought that this second step is inhibited in *norpA* mutants, leading to enhanced TIM degradation and an accumulation of CRY. Nevertheless, we were not able to reproduce the protecting effect of *norpA* loss-of-function on CRY (see response to point 7 of referee 1) and it is therefore no longer part of the revised manuscript. We now focus on the protective function of *Is-tim* instead.*

Minor comments:

Line 34: Gq-PLC- β needs to be explained.

Not relevant for the revised version

Line 43: something is missing in this sentence (perhaps: circadian clocks, which regulate)

Thank you, this sentence has been corrected

Line 46: 'terminate' instead of 'terminateMi'

Has been corrected

Line 52: I think that it is oversimplified to talk about CRY as the main circadian photoreceptor. This is even in contrast to the results of the corresponding author.

Has been changed to 'Cry is an important circadian photoreceptor'

Line 72: why not adding 'operating in the eyes' after 'cascade'. Although, the authors demonstrate later that PLC- β operates not only in the eyes, this would help understanding here.

Sentence has been removed from the revised version of the manuscript

Line 74: ')' is missing after '18'.

Reference has been removed from the revised version of the manuscript

Line 88: explain 'Gq'

Sentence has been removed from the revised version of the manuscript

Line 132 and Figure 1B: The entrainment of norpAp41 mutants with norpA rescued with tim-gal4 looks even better than that of wildtype flies. Is this because you overexpress norpA in the clock neurons, in which it is possibly expressed at low levels? Or, is this due to norpA expression in the eyes?

Sentence and Figure have been removed from the revised version of the manuscript

Lines 176: The sentence 'Introducing one copy of ls-tim is not sufficient...' disturbs the reading flow. I propose to move it. Just one sentence later, it appears adequate.

Sentence has been removed from the revised version of the manuscript

Line 179: Do not abbreviate 'EI' in the text. See also line 239.

EI has been removed from the revised version of the manuscript

Line 235: 'Gentile et al., 2013b' needs to be cited as a number.

This paper is now cited correctly

Line 249: Please explain 'rh1-4'

Sentence has been removed from the revised version of the manuscript

Line 252: '(add references)' need to be exchanged by real references.

Sentence has been removed from the revised version of the manuscript

Line 252: 'In contrast' does not fit. I would rather write 'Similarly'

Sentence has been removed from the revised version of the manuscript

Line 261: 'rescues' appears inappropriate. What about 'restores'?

Sentence has been removed from the revised version of the manuscript

Line 278: What about 'enabling' instead of 'regulating'?

Sentence has been removed from the revised version of the manuscript

The References need a thorough editing. In many places, the spaces between the words are missing. The abbreviated Journal names have sometimes a full stop after the abbreviations, sometimes not. Reference 34 lacks the Journal name and something is wrong with the author names. Species names are not italicized. In reference 44, *Drosophila* is written in lower case letters...

References have been checked in the revised version of the manuscript

Reviewer #3 (Remarks to the Author):

This is a very interesting ms from the Stanewsky lab which focuses on a long neglected but intriguing feature of fly chronobiology, namely how temperature cycles can overcome the arrhythmic behaviour induced by constant light (LL0). The ms is crisply written and the narrative flows very nicely, indeed I got quite excited wondering what the next section would bring. The ms focuses on the effects of *norpA* on temperature entrainment and how this interacts with CRY and TIM. The experiments also show the fundamental differences between the *ls-TIM* and *s-TIM* isoforms in temperature entrainment, which have so far been limited to circadian and photoperiodic light sensitivity. There is also an additional evolutionary context at the end of the Discussion, that will appeal to both neurogeneticists as well as behavioural biologists interested in evolution/population genetics. What the ms does not show is the molecular mechanism by which *norpA* performs its inhibitory role on CRY. Below

are my comments

*We thank this referee for acknowledging the importance of our work. Even though the mechanism of how *Drosophila melanogaster* can maintain rhythmic behavior and clock function in Northern latitudes is different from the one we originally proposed, we think it is equally important. In addition, the requirement of *ls-tim* for synchronizing to long summer days typical for Northern latitudes, supplies an additional (or alternative) reason for the directed selection of this novel *tim* allele.*

Fig S1. Please add arrows to the startle and the main activity component for the mutants as the differences in the mutants may not be obvious to a non specialist. The *norpA* main locomotor phenotype is advanced in LD, which is probably due to enhanced *per* splicing (refs 26,27). Might the larger advance of the mutant main activity in TC be due to even greater *per* splicing under these conditions which would carry over to the first days of DD? Refs 16,28 suggest no daily changes in *per* splicing under TC but my question is different – is the overall level of *per* splicing (irrespective of any daily changes) enhanced under TC where the average daily temperature is ~20°C?

Figure S1 is no longer part of the revised manuscript

L121 severe

The relevant sentence has been removed from the revised manuscript

L127 Fig 1A -the p24 mutant looks much more like WT than the p41 mutant

Figure is no longer part of the revised manuscript

Fig 2B the DN2s in yw do not show an oscillation, which is odd? Was the interaction term in the 2-way ANOVA significant? Could the asterisks be printed in a different colour to red and black as they are difficult to distinguish from the black datapoints?

The y w; s-tim data for TIM and PER for LLTC is now in Figure 2A and 2B of the revised version. The data for DDTC have not been included in the revised manuscript, because both s-tim and ls-tim flies robustly synchronize their behavior in DDTC (Figure S1C).

Fig 3A It's interesting that the mutants on ls-tim show a reduced startle to cold-hot, but increased startle to hot-cold transition...more sensitive to cold? reminds me of the splicing of per with norpA which means that the mutant behaves as if it is colder than it really is.

This is indeed interesting and we will address this potential effect of norpA on LLTC entrainment in the follow up paper focusing on norpA (see response to point 1 of referee 1).

L219-220 and Fig 4. Perhaps a little more guarded about ninaE mutants given the behavioral phenotypes are not as clear cut as in wild-type.

ninaE mutant behavior is no longer part of the revised manuscript. Instead we now show the more convincing gmr-hid; cry02 and norpA; cry02 behavior to demonstrate the role of the visual system for setting the behavioral activity phase in LLTC.

L227 Abruzzi et al PLoS Genet (2017) 13:e1006613 have shown norpA to be cycling in LNDs so it is expressed there as well. (You need to dig into the supplementaries of that paper to find it). Also 'recently' is inappropriate for ref 42 which is 10 years old!

The reviewer is correct, but the role of norpA expression in clock neurons for LLTC is no longer part of this study. Instead we show now that expression of ls-tim within clock neurons is sufficient for synchronization in LLTC (Figure 5).

L235 Gentile et al reference needs to be numbered 31

Gentile et al reference is now cited correctly

L252 'add references'?

fixed

L245-259 this could be compressed and figure 6 placed in supplementaries as it is a database RNAseq study and, although relevant, is the least interesting of the main results.

Results text and Figure no longer part of the revised manuscript

Fig 7C comment required on why TIM peaks in the cold compared to the warm in the norpA; cry mutants which restores a DDTC like profile?

Figure 7C is now Figure 4C. We think the referee refers to the difference between s-tim (peak between ZT6-9, Figure 2A, and 4C of the revised version) and norpA; cry (peak between ZT9-16, Figure 4C). We show in the revised version that although TIM reproducible shows a minor peak during the warm phase, TIM localization is constitutively cytoplasmic (Figure S3A, C). In the norpA; cry double mutants TIM cycles with a much higher amplitude and also is nuclear at the expected times (Figure 4C, S5 of the revised manuscript). The broad peak of expression extends from the warm to the cold phase. This, compared to s-tim and s-tim, cry02, advanced peak of TIM indeed correlates with a DDTC profile of locomotor behavior as the reviewer pointed out. In other words, if all light input is removed (visual system and Cry), flies in LLTC behave like wild type flies in DDTC. This is now included in the Discussion of the revised manuscript (p. 16).

Fig 8E comment required on why the double mutant shows less CRY under LD 25oC

Figure 8 is no longer part of the revised manuscript, because the stabilizing effect of norpA mutants on CRY turned out to be not reproducible (see response to point 7 of referee 1).

L336 Is-tim allele arose approx. 10,000 years ago. This has been amended more recently by Zonato et al (2018) J Biol Rhythms 33(1):15-23. The allele may have arisen, I believe between 300-3000 years ago

The Zonato et al reference has been added to the manuscript and the requirement of Is-tim for LLTC (Figure 1) synchronization and synchronization to white nights (Figure 6) is being discussed as driving force for the directed selection of this novel tim allele.

L341 D. melanogaster to my knowledge have not been found north of the Arctic circle but are found in southern to mid Scandinavia where summer photoperiods will be extremely long. If under these longer photoperiods the clock becomes less robust, but temperature cycles can restore some rhythmicity, this may be advantageous. However Helfrich-Forster has argue in a couple of recent papers that perhaps other more northern species of Drosophilids do not need to be rhythmic under these conditions. Perhaps a nod to this alternative view might be warranted. My personal feeling is that tropically evolved flies (like D. melanogaster) need to be rhythmic and will seek to maintain rhythmicity under extreme photoperiods, so the mechanism proposed here will help.

We agree with the referee and this aspect has become the main focus of the revised version. In fact, we are thankful for the comment about the long summer days in Scandinavia, which prompted us to perform the experiments under long day and temperature-cycling Oulu conditions. They showed that Is-tim is indeed required for synchronization to semi-natural conditions mimicking long summer days in Scandinavia.

A line or two about how norpA might mechanistically block CRY degradation would be welcomed. *Since we could not confirm the positive effect of norpA mutants on CRY stability, this possibility is no longer part of the revised manuscript.*

Reviewers' Comments:

Reviewer #1:

Remarks to the Author:

The new version of the paper focuses on the the s-tim ls-tim behavioral comparison which clearly shows that the natural ls-tim allele allows a robust peak of activity in constant light with temperature cycles (LLTC) whereas the s-tim allele fails to do so. The authors provide clock protein analysis data in the clock neurons, that show robust cycling in ls-tim but not in s-tim flies. Removing the Cryptochrome photoreceptor restores LLTC behavioral and molecular oscillations, supporting the hypothesis that the light hypersensitivity of s-tim flies is responsible for their non-ability to behave rhythmically in LLTC. Finally, the authors analyze the behavior of the genotypes in semi-natural conditions mimicking northern latitude and provide evidence that the ls-tim allele allows a much better behavioral synchronization with "day-night" cycles, supporting a selective advantage of ls-tim in these conditions.

Showing that a particular allele of timeless that make flies less sensitive to light allows temperature cycles to override the behavioral arrythmicity that is induced by constant light is clearly interesting and the experiments in semi-natural conditions provide a nice illustration of the advantage that the ls-tim allele could confer in northern latitudes. The results are globally convincing although some of the data need to be clarified as detailed below.

Major points:

1) Behavioral rhythmicity in LLTC (Figure 1AB) it is not easy to see whether the iso s-tim flies anticipate the temperature evening transition or not. The evening peak is of course much smaller than the one observed for iso31 ls-tim flies but on the actogram (A) they seem to show a small anticipatory activity every day, clearly starting before lights-off, and I cannot say whether they show anticipation on the B graph. It is not clear to me whether it is a problem with the graphic representation or not. If present, does the small anticipatory peak correspond to what the authors call "poor synchronization"? The absence of anticipation is clear in yw s-tim flies (Fig S1B), although the activity profiles are very different between iso s-tim and yw s-tim flies. The authors should clarify this, maybe by providing numbers (activity levels before the transition or some anticipatory index) and better defining what synchronization is in such conditions.

2) p6. "In contrast to most of the white-eyed s-tim flies we tested, the two red-eyed s-tim strains showed synchronised behaviour (compare Figure 1A, B and Figure S2A, B)."

The authors then observe that the red-eyes s-tim flies behave like timKO flies and thus conclude that red-eyed s-tim do not synchronize. I agree with the conclusion but the yw s-tim also had a similar profile (FigS1B) and it is thus difficult to conclude that the Tanzania and Houten s-tim flies have more daytime activity because of their red eyes.

I believe that this paragraph about red eyes brings some confusion with not much added value for the paper and it might be better to remove it.

3) Figure 2. The defect of TIM/PER oscillations in s-tim flies is clear. However:

- sLNvs do not seem to peak at ZT6 (thus with a 6h advance) as indicated in the text. Please clarify. When looking at the curves, it is difficult to see a stronger oscillation in sLNvs than in DN2s, which seem to have a peak at ZT6. Is the apparent oscillation in DN2s non-significant?
- I also do not understand why the authors claim that TIM levels are low in the DNS since they do not seem higher in the LNvs (particularly sLNvs). Please explain.

4) s-tim in the absence of CRY: as indicated by the authors, s-tim cry0 and ls-tim cry0 have a similar activity profile in LLTC (Fig 3AB). However, yw s-tim cryb show a strongly advanced evening peak. In fact, their evening peak does not seem to follow the new LLTC regime, as opposed to the ls-tim flies of the left panel. Is there synchronization with the TC cycle here? The phase in LLTC and LL seems similar to the one during the LD cycle. An advanced TC cycle could tell whether they really synchronize.

p9: "Overall, we observed robust PER and TIM oscillations in s-LNv and LNd clock neurons of s-tim cry02 flies...". I am not convinced that s-LNvs show robust oscillations, according to Figure 3D. Weak might be more appropriate here.

5) p11 and Fig 4. The behavioral data clearly show that the absence of the rhodopsin

photoreception induces a phase advance of the activity peak in LLTC. TIM oscillations are also clearly affected. However, I do not understand why the authors conclude that the phase advance is the result of the absence of *norpA*. The behavioral data in Fig4B showed a phase advance of *norpA* *s-tim cry0* compared to *s-tim cry0*, but the molecular data compare *s-tim* with *norpA s-tim cryb*. This experiment should compare *s-tim cry0* and *norpA s-tim cry0* like the behavioral experiment. See also the unexplained behavioral difference between *s-tim cry0* and *s-tim cryb*, as pointed in 4).

Minor points

"The observation that wild type flies carrying the ancient *s-tim* allele are not able to synchronise to LLTC demonstrate the advantage of the *ls-tim* allele in Northern latitudes."

I am not sure that the word "demonstrate" is appropriate here. "Support" would probably be better.

Reviewer #3:

Remarks to the Author:

This is a highly revised ms that bears little resemblance to the original. It focuses on an interesting subject that has been the subject of a number of high profile papers in the last few years, namely, how *Drosophilids* adapt to living at high latitudes. Some high latitude species reduce their light sensitivity by showing diminished expression of the blue light circadian photoreceptor, CRY, which produced some rhythmicity under constant light, as well as reduced expression of PDF, which generates arrhythmicity in constant darkness. These two molecular adaptations are useful in high latitudes because they make flies less light-sensitive and so maintain rhythmicity in Scandinavian summer (very long photoperiods) and in constant darkness (Arctic winters) when flies diapause, they can dispense with rhythmicity.

Tropical species such as *D. melanogaster* that have only relatively recently inhabited northern Europe. A recent mutation that originated in southern Europe 300-3000 years ago, *ls-tim*, has spread in every direction due to natural selection. It generates a less light sensitive form of TIM, the target of CRY. This leads to a less light sensitive circadian clock and higher levels of diapause, both characters that would be favoured in a seasonal photoperiodic environment such as Europe, as opposed to non-photoperiodic sub-Saharan Africa where *melanogaster* originated. However, when one thinks about northern European summers and the very long photoperiods (that never become darkness) that would generate arrhythmicity, why should flies maintain circadian rhythmicity under such 'arrhythmic' conditions? The answer lies in this paper by Stanewsky's group. The authors remind the reader that even under a natural Scandinavian summer, even though there is no real darkness, the temperature still cycles by ~10°C in every 24 h period, so *melanogaster* needs a circadian clock to entrain to these thermal rather than photoperiodic conditions.

The paper then goes through a logical sequence of experiments to show at behavioural and molecular levels, how *ls-tim* ALSO better adapts flies to rhythmicity under LL and temperature entrainment compared to the ancestral allele, *s-tim*. This new *ls-tim* phenotype can be observed under both laboratory and more natural simulated long-day (white-night) environmental condition. Furthermore, the *ls-tim* effect was believed (from previous work) to depend on TIM-CRY interactions. Here they show that in a *cry*-null background, *s-tim* also reveals the *ls-tim* temperature entrainment effect in LL, reinforcing the CRY-TIM interaction model. The *ls-tim* and *s-tim;cry*-null behavioural effects correlate with enhanced amplitude TIM cycling in some of the canonical clock neurons. These entrainment effects are also mediated partially by the photoreceptors (GMR-hid experiment) and the *norpA* encoded PLC- β phototransduction cascade. However replacing *ls-tim* in canonical clock neurons in a *tim*-null background is sufficient to restore the temperature entrainment effects but not in the photoreceptors.

This is an interesting story, the experiments are well done and it adds another important level of understanding to the *ls-tim* story, which represents one of the very few examples of a new adaptive mutation (non man-made such as insecticide resistance) sweeping through a population that has not yet gone to equilibrium.

I do have some comments the authors might wish to address

L 83 not 'directed' selection - 'directional'

The results are clearly presented, and in most cases it is obvious from eye-balling the data that

the effects the authors are seeing is real. In Fig 2 the authors might state on the legend that for Is-tim, all the points are significantly different from ZT6 if indeed they are?

Fig 3 – maybe a statistical test for the s-tim;cry02 v Is-tim;cry02 at LLTC6 might be helpful. I don't insist, but if there is a significant difference then it's OK to state that there is a partial rescue of the s-tim phenotype. If there is no sign diff then it looks like a full rescue. At least put some sems on the datapoints so the reader can judge

Fig 3C A general statement on the Figure legend of what is significantly different to what? Not so obvious on some of the plots.

Fig 4C – the 2 way ANOVA shows the orange different from the others. What about within genotype rhythmicity? This would come out in the posthoc tests from the same analyses. A mention in the legend is required.

L357 – the authors give the impression that D melanogaster are found north of the Arctic Circle. They may, but LL conditions are even found in Holland in midsummer (lat 54oN), because there is no darkness during the night – it is in a state of nautical or astronomical twilight between June and July so affair amount of light at night. The point is that the effects Lamaze et al describe for Is-tim are also adaptive much further south than Finland.

The authors might wish to speculate what would happen in say 50 or 100 years time. Would Is-tim become more dominant in the north of Europe, or might global warming reduce the need for a precocious diapause? That could be a way of untangling the advantageous effects of Is-tim on the clock and reproductive arrest.

REVIEWER COMMENTS

We thank both reviewers for the very constructive comments and suggestions for how we can improve the manuscript. In the revised version we paid special attention to improve the clarity of our Figures, which in several cases involved replacement with new Figures and data formats. In addition, we carefully revised our manuscript according to the comments and suggestions made by both referees. This involved a complete new PERIOD and TIMELESS staining experiment with the *norpA*[P41] *cry*[O2] double mutants, as requested by referee 1. We clearly marked all changes to the manuscript in RED font.

Reviewer #1 (Remarks to the Author):

The new version of the paper focuses on the the s-tim ls-tim behavioral comparison which clearly shows that the natural ls-tim allele allows a robust peak of activity in constant light with temperature cycles (LLTC) whereas the s-tim allele fails to do so. The authors provide clock protein analysis data in the clock neurons, that show robust cycling in ls-tim but not in s-tim flies. Removing the Cryptochrome photoreceptor restores LLTC behavioral and molecular oscillations, supporting the hypothesis that the light hypersensitivity of s-tim flies is responsible for their non-ability to behave rhythmically in LLTC. Finally, the authors analyze the behavior of the genotypes in semi-natural conditions mimicking northern latitude and provide evidence that the ls-tim allele allows a much better behavioral synchronization with “day-night” cycles, supporting a selective advantage of ls-tim in these conditions.

Showing that a particular allele of timeless that make flies less sensitive to light allows temperature cycles to override the behavioral arrhythmicity that is induced by constant light is clearly interesting and the experiments in semi-natural conditions provide a nice illustration of the advantage that the ls-tim allele could confer in northern latitudes. The results are globally convincing although some of the data need to be clarified as detailed below.

Major points:

1) Behavioral rhythmicity in LLTC (Figure 1AB) it is not easy to see whether the iso s-tim flies anticipate the temperature evening transition or not. The evening peak is of course much smaller than the one observed for iso31 ls-tim flies but on the actogram (A) they seem to show a small anticipatory activity every day, clearly starting before lights-off, and I cannot say whether they show anticipation on the B graph. It is not clear to me whether it is a problem with the graphic representation or not. If present, does the small anticipatory peak correspond to what the authors call “poor synchronization”? The absence of anticipation is clear in *yw* s-tim flies (Fig S1B), although the activity profiles are very different between iso s-tim and *yw* s-tim flies.

The authors should clarify this, maybe by providing numbers (activity levels before the transition or some anticipatory index) and better defining what synchronization is in such conditions.

We agree with the reviewer that it is difficult to distinguish anticipatory behavior from a plain startle response to the cold transition on the actograms and line graphs (previous Figures 1A, B). We therefore now also show histograms of the average activity (Figure 1B, S1B), which clearly demonstrate the lack of anticipation in s-tim flies and show that the activity increase occurs after the transition to the cold temperature (i.e. it is a startle response). In addition, we now supply quantification of the anticipatory behaviour by calculating the slope of the activity increase leading up to the synchronized evening activity peak in the second part of the warm phase (Figure 1C)

2) p6. “In contrast to most of the white-eyed s-tim flies we tested, the two red-eyed s-tim strains showed synchronised behaviour (compare Figure 1A, B and Figure S2A, B).”

The authors then observe that the red-eyes s-tim flies behave like timKO flies and thus conclude that red-eyed s-tim do not synchronize. I agree with the conclusion but the yw s-tim also had a similar profile (FigS1B) and it is thus difficult to conclude that the Tanzania and Houten s-tim flies have more daytime activity because of their red eyes.

I believe that this paragraph about red eyes brings some confusion with not much added value for the paper and it might be better to remove it.

We agree with the referee and removed this section as it does not add much to the message of the paper. We now say that s-tim flies behave similarly to tim^{KO} and therefore, the weak synchronization we can observe in some strains (as well as in timKO) may not be circadian-related. Nevertheless, we think it is important to show the behaviour of s-tim and ls-tim flies with various genetic backgrounds to demonstrate that behavioural differences are indeed due to the timeless polymorphism. The behaviour data of the various control strains is summarized in Figure S1 and S2 of the revised version of the manuscript.

3) Figure 2. The defect of TIM/PER oscillations in s-tim flies is clear. However:

- sLNvs do not seem to peak at ZT6 (thus with a 6h advance) as indicated in the text. Please clarify. When looking at the curves, it is difficult to see a stronger oscillation in sLNvs than in DN2s, which seem to have a peak at ZT6. Is the apparent oscillation in DN2s non-significant?

In the original version of the manuscript we reported a 6 hr advance of TIM oscillations and a reduced amplitude in s-tim compared to ls-tim flies in all Lateral Neurons and the DN1 and DN2. We now repeated these stainings and again observed low amplitude and 6 h phase advanced TIM oscillations in the LN_d, l-LN_v, 5ths-LN_v, and DN2 (oscillation is significant, statistics now added to revised Figure 2A), but not in the DN1. The s-LN_v showed fluctuating TIM levels, with a peak at ZT0 (compared to ZT12 for ls-tim) and we now indicate this in the revised text. Overall the phase advanced TIM oscillations are therefore reproducible and we attribute the minor differences between the original and current data to the very low amplitude of TIM cycling, which makes reliable quantification difficult. Importantly, we now show that sTIM is always cytoplasmic suggesting that despite the phase difference, TIM oscillations most likely do not reflect proper molecular clock synchronization. And indeed, using PER immunostaining (as requested by the reviewer) we found that PER was flat in all clock neurons studied in s-tim flies. We did notice a PER oscillation in the DN3, however, it is very weak and because TIM is not cycling in these neurons, it most likely does not reflect proper molecular clock synchronization in these neurons either.

- I also do not understand why the authors claim that TIM levels are low in the DNS since they do not seem higher in the LNvs (particularly sLNvs). Please explain.

We do say at the beginning of the paragraph that TIM levels are reduced in s-tim compared to ls-tim flies. What we meant here was that in the DN1 and DN3 the levels are constitutive, i.e., not cycling like in most of the LNs. We have changed the relevant passage in the results text to make this clearer: 'In the DN1 and DN3, S-Tim levels were at constant low levels at all four time points'

4) s-tim in the absence of CRY: as indicated by the authors, s-tim cry0 and ls-tim cry0 have a similar activity profile in LLTC (Fig 3AB). However, yw s-tim cryb show a strongly advanced evening peak. In fact, their evening peak does not seem to follow the new LLTC regime, as opposed to the ls-tim flies of the left panel. Is there synchronization with the TC cycle here? The phase in LLTC and LL seems

similar to the one during the LD cycle. An advanced TC cycle could tell whether they really synchronize.

We agree with the reviewer and did perform a phase advance experiment with $y w; s\text{-tim}; cryb$ flies. As you can see from the figure below, these flies can indeed synchronize to LLTC, confirming our conclusion that $s\text{-tim}$ can synchronize to LLTC when Cry function is impaired.

$s\text{-tim}; cry^b$ (n=16)

Figure 1: Behavior of $s\text{-tim} cry^b$ flies during LL constant temperature (days 1-3), LLTC (16°C:25°C) (days 4-10) and the same LLTC phase-advanced by 6 hr (days 11-16).

Nevertheless, for clarity reasons we decided to leave all $cryb$ data out of the revised version of the manuscript (behavior and immunostaining, see answer to point 5 below).

p9: "Overall, we observed robust PER and TIM oscillations in s-LNv and LN_d clock neurons of $s\text{-tim} cry02$ flies...". I am not convinced that s-LNvs show robust oscillations, according to Figure 3D. Weak might be more appropriate here.

We agree with the reviewer that PER oscillations in the s-LNv clock neurons (Figure D) are weak, but we now include statistics showing that they are significant. Moreover, in the same neurons TIM cycles with robust amplitude. Nevertheless, we have corrected the text accordingly to acknowledge the weak PER cycling in LNv clock cells (large and small).

5) p11 and Fig 4. The behavioral data clearly show that the absence of the rhodopsin photoreception induces a phase advance of the activity peak in LLTC. TIM oscillations are also clearly affected. However, I do not understand why the authors conclude that the phase advance is the result of the absence of $norpA$. The behavioral data in Fig4B showed a phase advance of $norpA s\text{-tim} cry0$ compared to $s\text{-tim} cry0$, but the molecular data compare $s\text{-tim}$ with $norpA s\text{-tim} cryb$. This experiment should compare $s\text{-tim} cry0$ and $norpA s\text{-tim} cry0$ like the behavioral experiment. See also the unexplained behavioral difference between $s\text{-tim} cry0$ and $s\text{-tim} cryb$, as pointed in 4).

We agree with the reviewer and have now performed the TIM staining experiments with norpA [P41]; s-tim; cry[02] flies and also included anti-PER. The results show an earlier phase of TIM increase in all LNd neurons and the 5th s-LNv in the double mutants (in cry02 mutants inly the CRY-negative LNd show this early rise). But overall there is not much of a difference between norpA cry double and cry single mutants, indicating that the behavioural phase difference caused by the absence or presence of norpA function is most likely not due to a different phase of molecular oscillations. The drastic phase advance of cry mutants we observe in the absence of a functional visual system (gmr-hid or norpA[P41]) therefore indicates a switch of the neuronal network.

Minor points

“The observation that wild type flies carrying the ancient s-tim allele are not able to synchronise to LLTC demonstrate the advantage of the ls-tim allele in Northern latitudes.”

I am not sure that the word “demonstrate” is appropriate here. “Support” would probably be better. *We agree with the reviewer and replaced ‘demonstrate’ by ‘support’-*

Reviewer #3 (Remarks to the Author):

This is a highly revised ms that bears little resemblance to the original. It focuses on an interesting subject that has been the subject of a number of high profile papers in the last few years, namely, how Drosophilids adapt to living at high latitudes. Some high latitude species reduce their light sensitivity by showing diminished expression of the blue light circadian photoreceptor, CRY, which produced some rhythmicity under constant light, as well as reduced expression of PDF, which generates arrhythmicity in constant darkness. These two molecular adaptations are useful in high latitudes because they make flies less light-sensitive and so maintain rhythmicity in Scandinavian summer (very long photoperiods) and in constant darkness (Arctic winters) when flies diapause, they can dispense with rhythmicity.

Tropical species such as *D. melanogaster* that have only relatively recently inhabited northern Europe. A recent mutation that originated in southern Europe 300-3000 years ago, *ls-tim*, has spread in every direction due to natural selection. It generates a less light sensitive form of TIM, the target of CRY. This leads to a less light sensitive circadian clock and higher levels of diapause, both characters that would be favoured in a seasonal photoperiodic environment such as Europe, as opposed to non-photoperiodic sub-Saharan Africa where *melanogaster* originated. However, when one thinks about northern European summers and the very long photoperiods (that never become darkness) that would generate arrhythmicity, why should flies maintain circadian rhythmicity under such ‘arrhythmic’ conditions? The answer lies in this paper by Stanewsky’s group. The authors remind the reader that even under a natural Scandinavian summer, even though there is no real darkness, the temperature still cycles by ~10oC in every 24 h period, so *melanogaster* needs a circadian clock to entrain to these thermal rather than photoperiodic conditions.

The paper then goes through a logical sequence of experiments to show at behavioural and molecular levels, how *ls-tim* ALSO better adapts flies to rhythmicity under LL and temperature entrainment compared to the ancestral allele, *s-tim*. This new *ls-tim* phenotype can be observed under both laboratory and more natural simulated long-day (white-night) environmental condition. Furthermore, the *ls-tim* effect was believed (from previous work) to depend on TIM-CRY interactions. Here they show that in a *cry*-null background, *s-tim* also reveals the *ls-tim* temperature entrainment

effect in LL, reinforcing the CRY-TIM interaction model. The *ls-tim* and *s-tim*;cry-null behavioural effects correlate with enhanced amplitude TIM cycling in some of the canonical clock neurons. These entrainment effects are also mediated partially by the photoreceptors (GMR-hid experiment) and the *norpA* encoded PLC- β phototransduction cascade. However replacing *ls-tim* in canonical clock neurons in a *tim*-null

background is sufficient to restore the temperature entrainment effects but not in the photoreceptors.

This is an interesting story, the experiments are well done and it adds another important level of understanding to the *ls-tim* story, which represents one of the very few examples of a new adaptive mutation (non man-made such as insecticide resistance) sweeping through a population that has not yet gone to equilibrium.

I do have some comments the authors might wish to address
L 83 not 'directed' selection – 'directional'

We have corrected this error

The results are clearly presented, and in most cases it is obvious from eye-balling the data that the effects the authors are seeing is real. In Fig 2 the authors might state on the legend that for *ls-tim*, all the points are significantly different from ZT6 if indeed they are?

*We agree with the referee and performed statistical tests showing that in *ls-tim* all time points are significantly different from ZT12 (peak for TIM and trough for PER) and we included this now in the Figure legend. The exceptions are the DN3 for TIM and the I-LNV for PER, for which we indicate significance or non-significance within the Figure 2.*

Fig 3 – maybe a statistical test for the *s-tim*;cry02 v *ls-tim*;cry02 at LLTC6 might be helpful. I don't insist, but if there is a significant difference then it's OK to state that there is a partial rescue of the *s-tim* phenotype. If there is no sign diff then it looks like a full rescue. At least put some sems on the datapoints so the reader can judge

*We have calculated the slope for both genotypes and while there is no significant difference on LL2. While the slope is positive for both *ls-tim* and *s-tim* in *cry02* background on LLTC6 the difference between the two genotypes just reaches significance level ($p < 0.05$) indicating a partial rescue of *s-tim* by *cry02*. We. These data are now shown in Figure S4A and we revised the results text accordingly:*

*'Also, the slope of the evening activity increase is positive for both *ls-tim* and *s-tim cry*⁰² flies, confirming synchronization to LLTC for both genotypes (Figure S4A). Nevertheless, the slightly lower slope value in *s-tim cry*⁰² flies on day 6 of LLTC ($p < 0.05$) indicates a partial rescue of *s-tim* by the removal of CRY functions, although there is no difference on day two in constant conditions after the TC (Figure S4A). Moreover, upon release into LL and constant temperature, activity peaks of both genotypes were aligned with those during the last few days in LLTC, indicating stable synchronisation of clock-driven behavioural rhythms (Figure 3A, B).*

Fig 3C A general statement on the Figure legend of what is significantly different to what? Not so obvious on some of the plots.

We agree with the referee and have now indicated significance within the Figure on some of the not so obvious plots

Fig 4C – the 2 way ANOVA shows the orange different from the others. What about within genotype rhythmicity? This would come out in the posthoc tests from the same analyses. A mention in the legend is required.

*These data have been replaced by a new staining experiment obtained with *norpA*[P41] *s-tim cry*[02] flies to allow better comparison with the behavioural data shown for the same genotype (Figure 4A). We also included statistical tests for rhythmicity within and between neuronal groups.*

L357 – the authors give the impression that *D melanogaster* are found north of the Arctic Circle. They may, but LL conditions are even found in Holland in midsummer (lat 54oN), because there is no darkness during the night – it is in a state of nautical or astronomical twilight between June and July so affair amount of light at night. The point is that the effects Lamaze et al describe for *Is-tim* are also adaptive much further south than Finland.

We thank the reviewer for making this point and we have changed the Discussion text accordingly:

*‘In addition to an earlier onset of winter, northern latitudes close to the Arctic Circle are also characterized by extremely long photoperiods in the summer. Because it never gets completely dark, even regions more South, for example the Northern regions of the Netherlands, Germany and Poland, as well as the Baltic states, experience considerable amounts of light at night’ in the summer. Our finding that *Is-tim* enables flies to synchronise to temperature cycles in constant light and particularly to semi-natural conditions mimicking white nights in Finland, indicates that this allele provides an additional fitness advantage during long photoperiods in central and Northern Europe*

The authors might wish to speculate what would happen in say 50 or 100 years time. Would *Is-tim* become more dominant in the north of Europe, or might global warming reduce the need for a precocious diapause? That could be a way of untangling the advantageous effects of *Is-tim* on the clock and reproductive arrest.

*We find it difficult to speculate on this. Even if global warming will reduce the advantage of entering diapause earlier, this could simply lead to an even further Northern distribution of *Drosophila*, perhaps even North of the Arctic Circle. In other words, in the face of global warming *Is-tim* would still be advantageous for both traits: earlier reproductive rest and ability to synchronize to temperature cycles in LL and support continuation of its Northern expansion.*

Reviewers' Comments:

Reviewer #1:

Remarks to the Author:

The authors have properly addressed all the points that I have raised on the previous version of the manuscript and I do not have more comment and/or suggestions.

Reviewer #3:

Remarks to the Author:

I'm pleased to see the authors have responded positively to my comments and the extra analyses and experiment add consistency to the story.

A minor comment

I.588 differed significantly from ZT12 (peak for TIM and trough for PER) apart and the I-LN_v (B): should be 'apart from the I-LN_v' I think

A more substantial comment

I. 369-371. I think its difficult to compare the fitness benefits of the white nights entrainment of Is-tim compared to the earlier diapause. Unless one does the proper experiments to show changes in the number of offspring produced in subsequent generations between s-tim and Is-tim (and these are difficult to do in a naturalistic manner), one really cannot state what the relative contributions of these two phenotypes are to the positive selection experienced by Is-tim. Perhaps a more appropriate sentence might be

'we propose that the ability to synchronise to temperature cycles in long summer days constitutes a major positive selection drive for this allele.'

Apart from these two comments, I congratulate the authors on an excellent piece of work that illuminates the evolution and population genetics of a clock gen within a neurogenetic context, something that is quite rare in chronobiology.

Response to Reviewers

Reviewer #1 (Remarks to the Author):

The authors have properly addressed all the points that I have raised on the previous version of the manuscript and I do not have more comment and/or suggestions.

Thank you!

Reviewer #3 (Remarks to the Author):

I'm pleased to see the authors have responded positively to my comments and the extra analyses and experiment add consistency to the story.

A minor comment

I.588 differed significantly from ZT12 (peak for TIM and trough for PER) apart and the I-LNv (B): should be 'apart from the I-LNv' I think

We thank the reviewer for spotting this error and we corrected it (p.27 top, 'from' highlighted in red)

A more substantial comment

I. 369-371. I think its difficult to compare the fitness benefits of the white nights entrainment of Is-tim compared to the earlier diapause. Unless one does the proper experiments to show changes in the number of offspring produced in subsequent generations between s-tim and Is-tim (and these are difficult to do in a naturalistic manner), one really cannot state what the relative contributions of these two phenotypes are to the positive selection experienced by Is-tim. Perhaps a more appropriate sentence might be

'we propose that the ability to synchronise to temperature cycles in long summer days constitutes a major positive selection drive for this allele.'

We agree with the referee and have changed the sentence according to this reviewers suggestion (page 17 top, 'major' highlighted in red)

Apart from these two comments, I congratulate the authors on an excellent piece of work that illuminates the evolution and population genetics of a clock gen within a neurogenetic context, something that is quite rare in chronobiology.

Thank you!